# CSF1R⁺ macrophage and osteoclast depletion impairs neural crest proliferation and craniofacial morphogenesis

Felix Ma[1,2], Rose Ru Jing Zhou[2,3], Matthew Rosin[2,4], Iris Zhou[2,3,*], Sabrina Ownsworth[2,3,*], Rouzbeh Ostadsharif Memar[1,2,5], Vincent B. Wong[2,6] and Jessica M. Rosin[2,4,‡]

## ABSTRACT

Despite a wealth of knowledge about the mechanisms underlying craniofacial morphogenesis during gestation, the roles of fetal macrophages and osteoclasts during this process remain less well characterized. Here, we used the pharmacological inhibitor PLX5622 to disrupt colony stimulating factor 1 receptor (CSF1R) signaling, which is essential for macrophage and osteoclast proliferation, differentiation and survival. Prenatal PLX5622 exposure in mouse resulted in ~50% depletion of CSF1R⁺ macrophages, with complete loss of osteoclasts. While there were no notable changes in craniofacial nerve or muscle development, prenatal exposure to PLX5622 resulted in skull doming and cranial suture impairments, in addition to disruptions to development of the premaxilla, mandible, ear ossicles, palate and cranial base. In response to PLX5622 exposure, cytokine and chemokine signaling was altered and neural crest proliferation was impaired. Our data also highlight sex- and strain-specific differences in PLX5622 phenotypes and together demonstrate that CSF1R⁺ macrophages and osteoclasts are essential for craniofacial morphogenesis.

KEY WORDS: Macrophage, Osteoclast, Neural crest, Embryogenesis, Craniofacial morphogenesis, Colony stimulating factor-1 receptor (CSF1R), Mouse

## INTRODUCTION

Colony stimulating factor 1 receptor (CSF1R) is expressed in cells of the mononuclear phagocyte lineage, namely microglia, macrophages and osteoclasts, and is required for their proliferation, differentiation and survival (Arai et al., 1999; Boulakirba et al., 2018; Marks et al., 1999; Giulian and Ingeman, 1988; Tushinski et al., 1982; Feng et al., 2002). CSF1R binds to two ligands, colony stimulating factor 1 (CSF1) and interleukin-34 (IL34), which stimulate proliferative and differentiating signals via STAT, AKT and ERK1/2 pathways (Boulakirba et al., 2018; Marks et al., 1999; Chihara et al., 2010). CSF1R is essential for phagocytic and bone resorptive functions in macrophages and osteoclasts, respectively (Feng et al., 2002; Delaney et al., 2021). Deficient CSF1R signaling in rodents has been shown to cause an osteopetrotic phenotype, where a lack of CSF1R⁺ cells leads to disruption of bone marrow cavities, loss of bone marrow hematopoiesis, and increased bone density (Dai et al., 2002; Marks and Lane, 1976; Pridans et al., 2018; Van Wesenbeeck et al., 2002). The *osteopetrotic* (*op*) phenotype has been characterized in *Csf1^op/op* ligand mutant mice and *Csf1r* knockout (KO) mice (Dai et al., 2002; Marks and Lane, 1976; Pollard et al., 1991; Erblich et al., 2011). These mice have reduced litter sizes and early postnatal lethality (up to 100%) depending on the strain (Dai et al., 2002; Marks and Lane, 1976; Erblich et al., 2011). Phenotypes observed in both *Csf1-* and *Csf1r*-disrupted mouse models also include a domed skull, shortened snout and toothlessness, while an osteosclerotic otic capsule, incus and stapes have only been observed in *Csf1^op/op* mice (Marks and Lane, 1976; Okano and Kishimoto, 2019; Sundquist et al., 1995). At the cellular level, these mice are deficient in blood monocytes and macrophages in most tissues; microglia are depleted in the receptor knockout but unaffected in the ligand mutant (Dai et al., 2002; Erblich et al., 2011; Begg et al., 1993; Cecchini et al., 1994; Wiktor-Jedrzejczak et al., 1982). The *Csf1 toothless* (*tl*) rat and *Csf1r* KO rat closely replicate the mouse phenotypes, including osteopetrosis, doming of the skull, a shortened snout, toothlessness, and microglia and macrophage depletion across most tissues (Pridans et al., 2018; Van Wesenbeeck et al., 2002). Interestingly, *Csf1r* KO rats also present with additional phenotypes, such as hypomineralization of the calvaria, impairment of the cranial sutures, bulging eyes and sclerosis of the cranial base (Pridans et al., 2018; Hume et al., 2020; Keshvari et al., 2021; Patkar et al., 2021). Viability is markedly improved in rat models, with only 23% of males dying postnatally and all females surviving beyond weaning (Pridans et al., 2018). In humans, bi-allelic *CSF1R* variants can cause bone phenotypes, such as osteosclerosis of the cranial vault and cranial base, in addition to narrowing of the optic canal (Guo et al., 2019; Monies et al., 2017).

To study the role of CSF1R⁺ cells in mediating the above-mentioned phenotypes, we previously reported a pharmacological inhibition model for disrupting these cells in a temporally controlled manner, focusing on embryogenesis (Rosin et al., 2018; Nagra et al., 2023). The CSF1R inhibitor PLX5622 was fed via chow to pregnant CD1 mice from embryonic day (E) 3.5 onwards to expose mouse embryos to the inhibitor (Rosin et al., 2018). Indeed, like the severe depletion of microglia in the *Csf1r* KO mouse, treatment with PLX5622 depleted ~99% of microglia in the hypothalamus by E15.5, demonstrating excellent oral bioavailability, penetrance and efficacy of this model (Rosin et al., 2018). Outside of the brain,

[1]Craniofacial Science Graduate Program, Faculty of Dentistry, The University of British Columbia, Vancouver, BC V6T 1Z3, Canada. [2]Life Sciences Institute, The University of British Columbia, Vancouver, BC V6T 1Z3, Canada. [3]Doctor of Dental Medicine Program, Faculty of Dentistry, The University of British Columbia, Vancouver, BC V6T 1Z3, Canada. [4]Department of Oral Biological and Medical Sciences, Faculty of Dentistry, The University of British Columbia, Vancouver, BC V6T 1Z3, Canada. [5]Division of Periodontics and Dental Hygiene, Faculty of Dentistry, The University of British Columbia, Vancouver, BC V6T 1Z3, Canada. [6]Neuroscience Program, Faculty of Science, McGill University, Montreal, QC H3A 1A1, Canada.

*These authors contributed equally to this work

‡Author for correspondence ( jessica.rosin@ubc.ca)

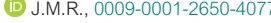 J.M.R., 0009-0001-2650-4077

PLX5622 exposure was found to cause a noticeable reduction in CSF1R⁺ cells throughout craniofacial tissues at E18.5 (Nagra et al., 2023). As in the mutant mouse models described, PLX5622 treatment also reduced litter sizes and increased postnatal lethality (~23%) (Rosin et al., 2018). These CD1 mice also presented with domed skulls in addition to morphological disruptions to the incisors and molars, although the continuous eruption of incisors did not appear to be inhibited (Rosin et al., 2018). Using geometric morphometric analyses, craniofacial and mandibular shape and size were shown to be significantly changed and decreased, respectively, by PLX5622 exposure (Nagra et al., 2023). Interestingly, a recent study showed that intragastric administration of PLX5622 to pregnant ICR mice between E14.5 and E17.5 reduced osteoclasts and resulted in an enlarged cartilaginous zone in the midpalatal suture (Yongzhen et al., 2024). However, previous studies did not fully characterize the depletion of CSF1R⁺ cells outside of the brain or across embryogenesis, nor did they thoroughly describe all craniofacial phenotypes or identify the underlying morphogenetic disruptions leading to the phenotypes observed using this pharmacological inhibition model.

In this study, we used our previously established pharmacological inhibition model to study the impact of prenatal PLX5622 exposure on craniofacial nerve, muscle, cartilage and bone development across embryogenesis in two strains (CD1 and C57BL/6) and both sexes (males and females). Exposure to PLX5622 resulted in a continuous and stable depletion of CSF1R⁺ macrophages to ~50% across gestation, in addition to the complete loss of osteoclasts. Although we did not observe overt changes in craniofacial nerve or muscle development across embryogenesis, disrupted bone phenotypes in craniofacial tissue of PLX5622-exposed animals could be observed as early as E15.5, with notable abnormalities in skull shape, cranial sutures, ear ossicles, premaxilla, mandible, palate and cranial base observed by birth. We identified significant alterations in the secretion of 16 cytokines and chemokines from cultured craniofacial tissues in response to prenatal PLX5622 exposure and validated that several such secreted factors are indeed expressed by CSF1R⁺ macrophages. At the cellular level, exposure to PLX5622 across gestation decreased neural crest cell (NCC) proliferation both *in vitro* in a sphere assay and *in vivo* in embryonic craniofacial tissue cryosections, resulting in fewer NCCs in tissues impacted by PLX5622 exposure. Together, these data demonstrate that CSF1R⁺ macrophages and osteoclasts are required for craniofacial morphogenesis during the embryonic period and suggest that altered signaling to NCCs may underlie the phenotypes observed in PLX5622 offspring.

## RESULTS

### Embryonic exposure to the CSF1R inhibitor PLX5622 depletes CSF1R⁺ macrophages and osteoclasts

As CSF1R function is essential for macrophage and osteoclast proliferation, differentiation and survival (Tushinski et al., 1982; Feng et al., 2002), we tested the impact of PLX5622 on reducing macrophage and osteoclast populations in craniofacial tissues across embryogenesis. Pregnant dams were fed control or PLX5622 diet starting at E3.5 to avoid disrupting embryo implantation, a process regulated by endometrial macrophages (Care et al., 2013; Wang et al., 2016), which was supported by our findings that administering PLX5622 gestationally did not significantly impact embryonic litter size (Table S1; CD1 $P=0.0698$, C57BL/6 $P=0.5683$), the number of embryo resorptions per litter (Table S1; CD1 $P=0.6110$, C57BL/6 $P=0.4176$), or embryo sex ratios (Table S1; CD1 $P=0.5676$, C57BL/6 $P=0.4612$) in CD1 or C57BL/6 mice. Craniofacial tissues were collected from $Csf1r^{EGFP}$ transgenic embryos, enabling CSF1R-expressing cells to be assessed by enhanced green fluorescent protein

(EGFP) signal using flow cytometry (Fig. 1A), which revealed a significant decrease in $Csf1r^{EGFP+}$ cells in response to PLX5622 exposure across all time points assessed (Fig. 1B; main effect of diet: E11.5 $F_{(1,18)}=24.42$, $P=0.0001$; E13.5 $F_{(1,24)}=29.53$, $P<0.0001$; E15.5 $F_{(1,17)}=27.25$, $P<0.0001$; E17.5 $F_{(1,16)}=29.45$, $P<0.0001$). Sustained depletion (40-63%) of $Csf1r^{EGFP+}$ cells was observed across gestation in PLX5622 male embryos, while a significant reduction in $Csf1r^{EGFP+}$ cells was only achieved at E17.5 for PLX5622 female embryos (Fig. 1B; E11.5 male $P=0.0009$, female $P=0.1321$; E13.5 male $P<0.0001$, female $P=0.0806$; E15.5 male $P<0.0001$, female $P=0.2610$; E17.5 male $P=0.0452$, female $P=0.0011$). Interestingly, significantly more $Csf1r^{EGFP+}$ cells were found in control males compared to females at E11.5 (Fig. 1B; significant impact of sex $F_{(1,18)}=5.801$, $P=0.0270$; control male versus female $P=0.0259$).

To visualize and quantify tissue-specific depletion of CSF1R-expressing cells within the embryo, craniofacial tissue cryosections were collected from E15.5 male and female $Csf1r^{EGFP}$ transgenic embryos and stained with a CSF1R antibody, as only ~92% of $Csf1r^{EGFP+}$ cells were found to also express CSF1R (Table S2). Quantification of CSF1R-expressing cells in and around several developing craniofacial structures of E15.5 PLX5622 embryos compared to controls revealed a significant decrease of both $Csf1r^{EGFP+}$ (Fig. 1C-W; main effect of diet: nasal septum $F_{(1,8)}=32.85$, $P=0.0004$; Meckel's cartilage $F_{(1,8)}=95.02$, $P<0.0001$; ear $F_{(1,8)}=74.37$, $P<0.0001$; eye $F_{(1,8)}=25.33$, $P=0.0010$; trigeminal $F_{(1,8)}=20.77$, $P=0.0019$; tongue $F_{(1,8)}=35.67$, $P=0.0003$) and CSF1R⁺ (Fig. 1C-W; main effect of diet: nasal septum $F_{(1,8)}=33.27$, $P=0.0004$; Meckel's cartilage $F_{(1,8)}=33.49$, $P=0.0004$; ear $F_{(1,8)}=112.6$, $P<0.0001$; eye $F_{(1,8)}=30.17$, $P=0.0006$; trigeminal $F_{(1,8)}=16.12$, $P=0.0039$; tongue $F_{(1,8)}=56.38$, $P<0.0001$) cells in all tissues examined except for the maxillary incisor (Fig. 1N). Specifically, CSF1R-expressing cells were significantly depleted in and around developing cartilaginous and/or bony structures of PLX5622 embryos, with only 53.7% of $Csf1r^{EGFP+}$ and 32.9% of CSF1R⁺ cells remaining around the nasal septum (Fig. 1C-E; $Csf1r^{EGFP}$ male $P=0.0044$, female $P=0.0601$; CSF1R male $P=0.0059$, female $P=0.0407$), 36.5% of $Csf1r^{EGFP+}$ and 23.8% of CSF1R⁺ cells remaining around Meckel's cartilage (Fig. 1F-H; $Csf1r^{EGFP}$ male $P=0.0006$, female $P=0.0005$; CSF1R male $P=0.0116$, female $P=0.0188$), and 32.1% of $Csf1r^{EGFP+}$ and 21.8% of CSF1R+ cells remaining in the ear (Fig. 1I-K; $Csf1r^{EGFP}$ male $P=0.0006$, female $P=0.0029$; CSF1R male $P=0.0002$, female $P=0.0007$). CSF1R-expressing cells were also significantly reduced in craniofacial nervous tissue of PLX5622 embryos, with only 39.8% of $Csf1r^{EGFP+}$ and 21.6% of CSF1R⁺ cells remaining in the eye (Fig. 1O-Q; $Csf1r^{EGFP}$ male $P=0.0205$, female $P=0.0453$; CSF1R male $P=0.0195$, female $P=0.0195$), and 56.7% of $Csf1r^{EGFP+}$ cells remaining in the trigeminal; with only male embryos displaying a significant change in the trigeminal (Fig. 1R-T; $Csf1r^{EGFP}$ male $P=0.0349$, female $P=0.0679$). Moreover, CSF1R-expressing cells were also significantly reduced in muscle tissue, with only 49.2% of $Csf1r^{EGFP+}$ and 27.5% of CSF1R⁺ cells remaining in the tongue of E15.5 PLX5622 embryos compared to controls (Fig. 1U-W; $Csf1r^{EGFP}$ male $P=0.0052$, female $P=0.0311$; CSF1R male $P=0.0016$, female $P=0.0065$).

While visualizing E15.5 $Csf1r^{EGFP}$ craniofacial tissue sections using high-resolution confocal microscopy, we observed multinucleated $Csf1r^{EGFP+}$ cells within several craniofacial bones such as the premaxilla, maxilla and mandible (Fig. 2A-F′, arrows). To quantify osteoclast-specific depletion following gestational exposure to PLX5622, we labeled osteoclasts with a cathepsin K

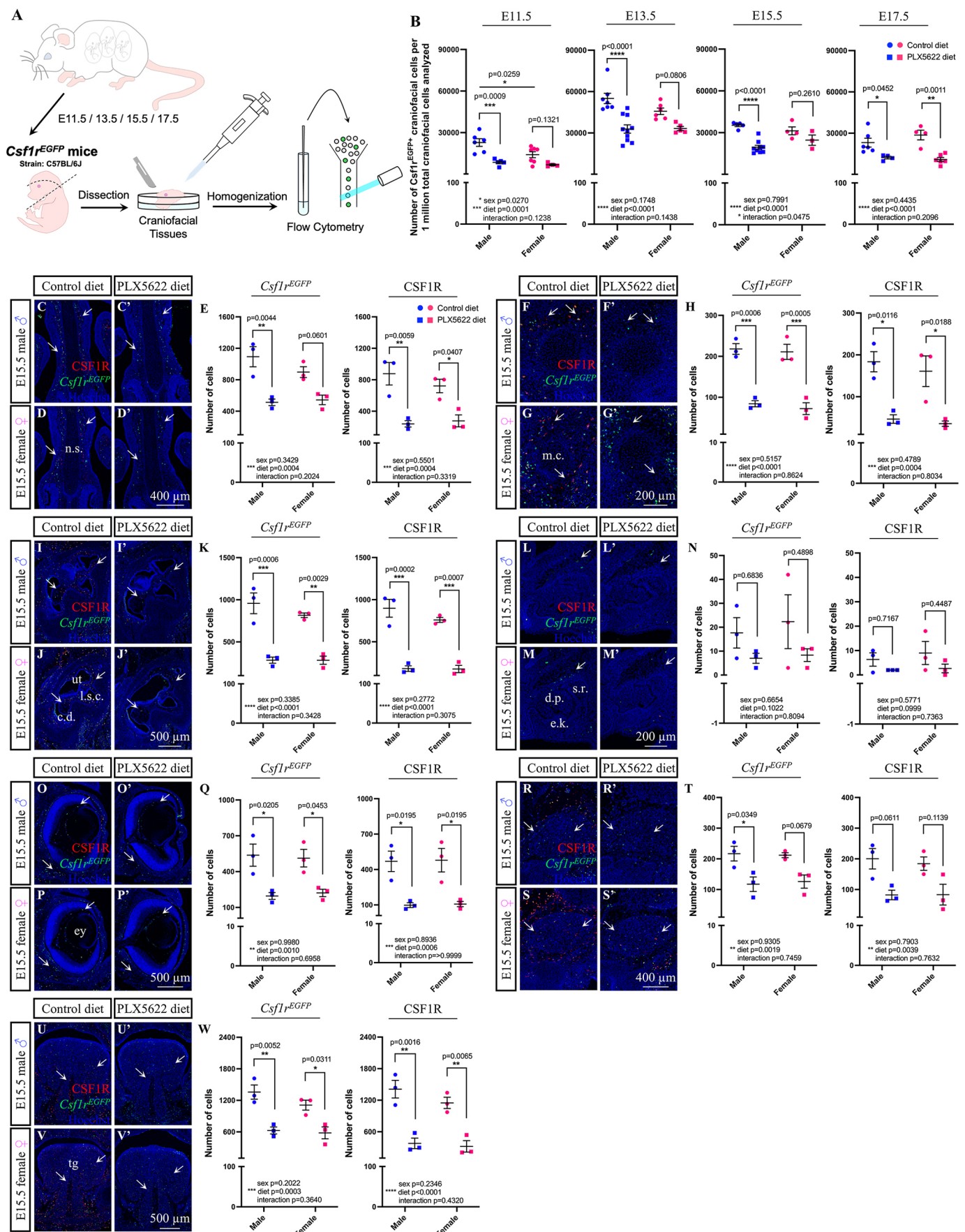

**Fig. 1.** See next page for legend.

**Fig. 1. Gestational exposure to PLX5622 significantly depletes CSF1R-expressing cells.** (A) Schematic illustrating collection of *Csf1r*$^{EGFP+}$ cells for flow cytometry. (B) Quantification of EGFP$^+$ cells from *Csf1r*$^{EGFP}$ craniofacial tissues (*n*=3-10 embryos per sex/treatment/time-point from two to four dams). (C-W) Immunofluorescence images and quantification of CSF1R$^+$ and *Csf1r*$^{EGFP+}$ cells in and around the E15.5 nasal septum (C-E), Meckel's cartilage (F-H), ear (I-K), maxillary incisor (L-N), eye (O-Q), trigeminal (R-T) and tongue (U-W). Arrows mark CSF1R and *Csf1r*$^{EGFP}$ double-positive cells (*n*=3 embryos per sex/treatment from two or three dams). c.d., cochlear duct; d.p., dental papilla; e.k., enamel knot; ey, eye; l.s.c., lateral semicircular canal; m.c., Meckel's cartilage; n.s., nasal septum; s.r., stellate reticulum; tg, tongue; ut, utricle. Blue dots represent male and pink dots represent female. Counts represent mean±s.e.m. and were analyzed by a two-way ANOVA with Tukey's post-hoc test.

(CTSK) antibody. Intriguingly, exposure to PLX5622 completely depleted CTSK$^+$/*Csf1r*$^{EGFP+}$ double-positive multinucleated osteoclasts in the E15.5 male and female mandible (Fig. 2E-I, arrows; main effect of diet $F_{(1,8)}$=49.85, *P*=0.0001; male *P*=0.0009, female *P*=0.0052). Moreover, there was a complete loss of all single-positive *Csf1r*$^{EGFP+}$ multinucleated cells in the E15.5 PLX5622 mandible compared to control for both sexes (Fig. 2E-I′; main effect of diet $F_{(1,8)}$=49.85, *P*=0.0001; male *P*=0.0009, female *P*=0.0052). As embryonic osteoclasts appeared to be absent in the E15.5 mandible of PLX5622 embryos compared to controls, we then looked to determine whether this led to loss of bone resorptive function across the embryonic head. Tartrate-resistant acid phosphatase (TRAP) staining of E15.5 craniofacial tissues revealed loss of osteoclastic activity surrounding developing craniofacial bones, including the premaxilla, mandible, maxilla, frontal and basioccipital bones (Fig. 2J-S′, arrows). Moreover, quantification of the TRAP$^+$ area in the premaxilla, mandible and maxilla revealed a significant reduction in TRAP staining in both E15.5 male and female embryos in response to PLX5622 exposure (Fig. 2T-V; main effect of diet: premaxilla $F_{(1,8)}$=101.0, *P*<0.0001; male *P*=0.0004, female *P*=0.0006; mandible $F_{(1,8)}$=65.09, *P*<0.0001; male *P*=0.0016, female *P*=0.0027; maxilla $F_{(1,8)}$=94.02, *P*<0.0001; male *P*=0.0004, female *P*=0.0010). Together, the data demonstrate that exposure to the CSF1R inhibitor PLX5622 during gestation significantly depletes CSF1R-expressing cells across embryogenesis and disrupts osteoclast development and activity.

## CSF1R$^+$ cell depletion results in tissue-specific changes in apoptosis in the developing embryo

Considering that one of the main functions of macrophages is to phagocytose apoptotic cells and cellular debris (Metschnikoff, 1891; van Furth et al., 1972), and apoptosis is an essential process in embryonic craniofacial development (Bourez et al., 1997; Dupe et al., 1999; Tang et al., 2005; Zakeri et al., 1994), craniofacial tissue cryosections collected from E15.5 *Csf1r*$^{EGFP}$ transgenic embryos were stained with an active cleaved caspase 3 (CC3) antibody to visualize whether apoptotic cells accumulated following CSF1R$^+$ cell depletion. We were also interested in whether we would observe CC3$^+$/*Csf1r*$^{EGFP+}$ double-positive cells, as the reduction in CSF1R$^+$ cells should be mediated in part by cell death. Quantification of CC3$^+$ single-positive and CC3$^+$/*Csf1r*$^{EGFP+}$ double-positive cells in and around several developing craniofacial structures of E15.5 embryos (Fig. S1A-U′, arrows) revealed a significant increase in apoptosis in the ear (Fig. S1I, $F_{(1,8)}$=12.06, *P*=0.0084; Fig. S1I′, $F_{(1,8)}$=7.043, *P*=0.0291), eye (Fig. S1O, $F_{(1,8)}$=9.904, *P*=0.0137; Fig. S1O′, $F_{(1,8)}$=6.601, *P*=0.0392), trigeminal (Fig. S1R, $F_{(1,8)}$=16.18, *P*=0.0038) and tongue (Fig. S1U, $F_{(1,8)}$=332.0, *P*<0.0001) in response to PLX5622 exposure, while no changes in apoptosis

were observed in and around the nasal septum (Fig. S1C,C′), Meckel's cartilage (Fig. S1F,F′) or the maxillary incisor (Fig. S1L,L′). A significant increase in CC3$^+$ single-positive cells was only observed in the E15.5 PLX5622 female ear (Fig. S1I; male *P*=0.8656, female *P*=0.0138), while no change in CC3$^+$/*Csf1r*$^{EGFP+}$ double-positive cells was observed in either sex (Fig. S1I′; male *P*=0.3976, female *P*=0.2363). Sex differences were also identified in the ear (Fig. S1I; significant impact of sex $F_{(1,8)}$=5.68, *P*=0.0443 and sex×diet interaction $F_{(1,8)}$=5.68, *P*=0.0443), whereby significantly more CC3$^+$ cells were observed in PLX5622 females compared to males (Fig. S1I; *P*=0.0395). Surprisingly, we did not observe a statistically significant change in CC3$^+$ single-positive (Fig. S1O; male *P*=0.2945, female *P*=0.1279) or CC3$^+$/*Csf1r*$^{EGFP+}$ double-positive (Fig. S1O′; male *P*=0.2944, female *P*=0.4463) cells in the eye for either sex. In contrast, CC3$^+$ cells showed a significant increase in the trigeminal of PLX5622 male embryos but not females (Fig. S1R; male *P*=0.0216, female *P*=0.3065). Intriguingly, the tongue was the only craniofacial structure assessed that displayed a significant increase in CC3$^+$ cells in both PLX5622 male and female embryos (Fig. S1U; male *P*<0.0001, female *P*<0.0001), in addition to sex differences (Fig. S1U; significant impact of sex $F_{(1,8)}$=7.364, *P*=0.0265), where there were significantly more CC3$^+$ cells observed in PLX5622 males compared to females (Fig. S1U; *P*=0.0406). Collectively, the data suggest that depleting CSF1R$^+$ cells during embryogenesis leads to an accumulation of apoptotic cells predominantly in nervous and muscular structures, but not in or around developing cartilaginous/bony structures, except for the ear.

## Embryonic exposure to the CSF1R inhibitor PLX5622 does not impact craniofacial nerve or muscle development

In this study, we sought to thoroughly characterize the craniofacial phenotypes resulting from embryonic PLX5622 exposure, especially as embryonic soft tissues were previously unexplored in *Csf1*/*Csf1r*-disrupted genetic models (Dai et al., 2002; Marks and Lane, 1976; Pridans et al., 2018; van Wesenbeeck et al., 2002; Erblich et al., 2011; Okano and Kishimoto, 2019; Keshvari et al., 2021) and our own pharmacological model using PLX5622 (Rosin et al., 2018; Nagra et al., 2023). Accordingly, we first examined craniofacial nerve development from E11.5 to E13.5 using a neurofilament antibody (2H3) and whole-mount immunostaining (Fig. S2A-L). Neurofilament staining in the trigeminal and facial nerves, including the zygomatic, buccal and marginal branches of the trigeminal, were comparable between E11.5 control and PLX5622 male and female CD1 (Fig. S2A-D) embryos. Similarly, neurofilament staining in the facial nerve, auriculotemporal nerve and great auricular nerve were comparable between E12.5 control and PLX5622 male and female CD1 (Fig. S2E-H) embryos. Moreover, craniofacial nerve development continued to be comparable at E13.5, whereby the great auricular nerve, auriculotemporal nerve, zygomatic branch of the trigeminal, superior buccolabial nerve, inferior buccolabial nerve and marginal mandibular nerve appeared similar between control and PLX5622 male and female CD1 (Fig. S2I-L) embryos.

Next, we examined craniofacial muscle development from E11.5 to E13.5 using a myosin heavy chain antibody (MF20) and whole-mount immunostaining (Fig. S2M-X). Muscle staining in the masseter, auricularis, buccinator and zygomaticomandibularis, appeared comparable across E11.5 to E13.5 in control and PLX5622 male and female CD1 embryos (Fig. S2M-X). To assess the development of muscles deeper in the tissues, immunofluorescence staining was performed on E15.5 craniofacial tissue cryosections using the MF20 antibody (Fig. S3). Muscle fiber diameter and density

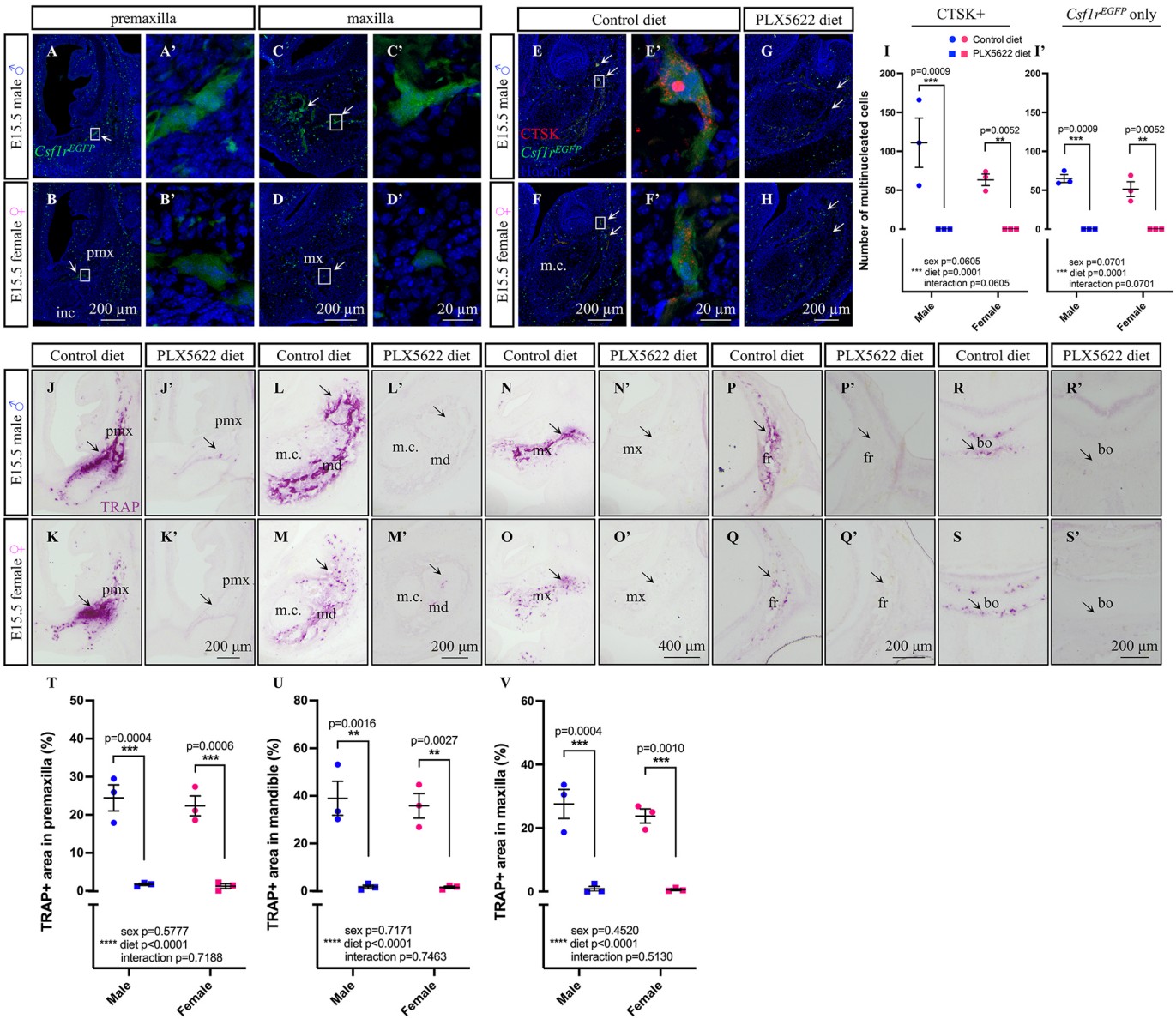

**Fig. 2. Prenatal exposure to PLX5622 disrupts osteoclast development and function.** (A-D′) $Csf1r^{EGFP+}$ osteoclasts (arrows) in the E15.5 premaxilla (A-B′) and maxilla (C-D′). (E-I) Immunofluorescence images (E-H) and quantification (I,I′) of multinucleated CTSK/$Csf1r^{EGFP}$ double-positive osteoclasts (arrows) (I) and $Csf1r^{EGFP+}$ single-positive osteoclasts (I′) in the E15.5 mandible. (J-S′) TRAP staining (arrows) in E15.5 premaxilla (J-K′), mandible (L-M′), maxilla (N-O′), frontal (P-Q′) and basioccipital (R-S′) bones. (T-V) Quantification of total TRAP$^+$ area in premaxilla (T), mandible (U) and maxilla (V). $n$=3 embryos per sex/treatment from two or three dams. bo, basioccipital; fr, frontal; m.c, Meckel's cartilage; md, mandible; mx, maxilla pmx, premaxilla. Counts represent mean±s.e.m. and were analyzed by an ART ANOVA (I,I′) or a two-way ANOVA (T-V) with Tukey's post-hoc test. A′-F′ show magnifications of the respective boxed areas in A-F.

appeared comparable in the buccinator, zygomaticomandibularis, deep masseter, superficial masseter, temporalis, pterygoid, tongue, and extraocular muscles, including the superior oblique, inferior oblique, superior rectus, medial rectus, inferior rectus and retractor bulbi, between E15.5 control and PLX5622 male and female C57BL/6 embryos (Fig. S3). These data suggests that depleting CSF1R$^+$ cells during embryogenesis does not noticeably impact the gross morphology of craniofacial nerves or muscles in male or female CD1 or C57BL/6 embryos.

## Prenatal exposure to the CSF1R inhibitor PLX5622 directs tissue- and time-dependent changes in craniofacial morphogenesis during embryogenesis

Previously, we reported doming of the cranial vault, mandibular disruptions, and dental phenotypes in postnatal day (P) 21 and P28

CD1 mice exposed to PLX5622 embryonically (Rosin et al., 2018; Nagra et al., 2023). However, our prior analyses did not fully characterize all the cartilaginous and/or bony phenotypes present in PLX5622 offspring. Moreover, the $Csf1/Csf1r$-disrupted genetic models and our own pharmacological PLX5622 studies failed to examine craniofacial morphogenesis during the embryonic period (Dai et al., 2002; Marks and Lane, 1976; Pridans et al., 2018; van Wesenbeeck et al., 2002; Okano and Kishimoto, 2019; Sundquist et al., 1995; Keshvari et al., 2021; Rosin et al., 2018; Nagra et al., 2023). As we observed a loss of osteoclasts and bone resorptive activity in E15.5 PLX5622 embryos in the current study (see Fig. 2), we utilized E15.5 craniofacial tissue cryosections to visualize the impact of PLX5622 exposure on osteoblasts and bone formation during embryogenesis. To start, immunofluorescence staining with an Sp7 antibody was performed to label osteoblasts. Disrupted

patterning in the premaxilla and mandible (Fig. 3A-D′, arrows) was observed in E15.5 male and female PLX5622 embryos, while other craniofacial bones such as the maxilla, frontal bone and basioccipital bone displayed comparable Sp7 staining (Fig. 3E-J′).

Next, we evaluated cartilage and bone development using Alcian Blue and von Kossa staining, respectively. Craniofacial cartilage development appeared unimpacted, as Alcian Blue staining was comparable between E15.5 control and PLX5622 male and female embryos (Fig. 3K-T′). In contrast, increased bone density in the premaxilla and mandible (Fig. 3K-N′, arrows) was observed in E15.5 male and female PLX5622 embryos, which complemented the altered osteoblast patterning observed in these bones. Moreover, consistent with Sp7 osteoblast staining, bone density in other craniofacial bones such as the maxilla, frontal bone and basioccipital bone (Fig. 3O-T′) appeared comparable between E15.5 control and PLX5622 male and female embryos. Quantification of unmineralized areas in the premaxilla and mandible, which appeared to fill in with

von Kossa staining in E15.5 PLX5622 embryos (Fig. 3K-N′, arrows), was shown to significantly decrease in response to PLX5622 exposure (Fig. 3U,V; main effect of diet: premaxilla $F_{(1,8)}=26.23$, $P=0.0009$; mandible $F_{(1,8)}=25.34$, $P=0.0010$). Although the unmineralized area significantly decreased in the mandible of both sexes (Fig. 3V; male $P=0.0319$, female $P=0.0289$), it was only found to be statistically significant in the premaxilla of E15.5 male embryos (Fig. 3U; male $P=0.0108$, female $P=0.0749$). Importantly, quantification of mineralization in the maxilla, a region that displayed comparable von Kossa staining between E15.5 control and PLX5622 embryos (Fig. 3O-P′), showed no change in mineralization in response to PLX5622 exposure (Fig. 3W).

To investigate whether the changes in osteoblast/bone development in PLX5622 embryos begin prior to E15.5, micromass cultures of mesenchymal cells collected from the frontonasal mass (which gives rise to premaxilla) (Iyyanar et al., 2023; Richman and Tickle, 1989; Wedden, 1987) and mandible of E12.5 control and PLX5622 male

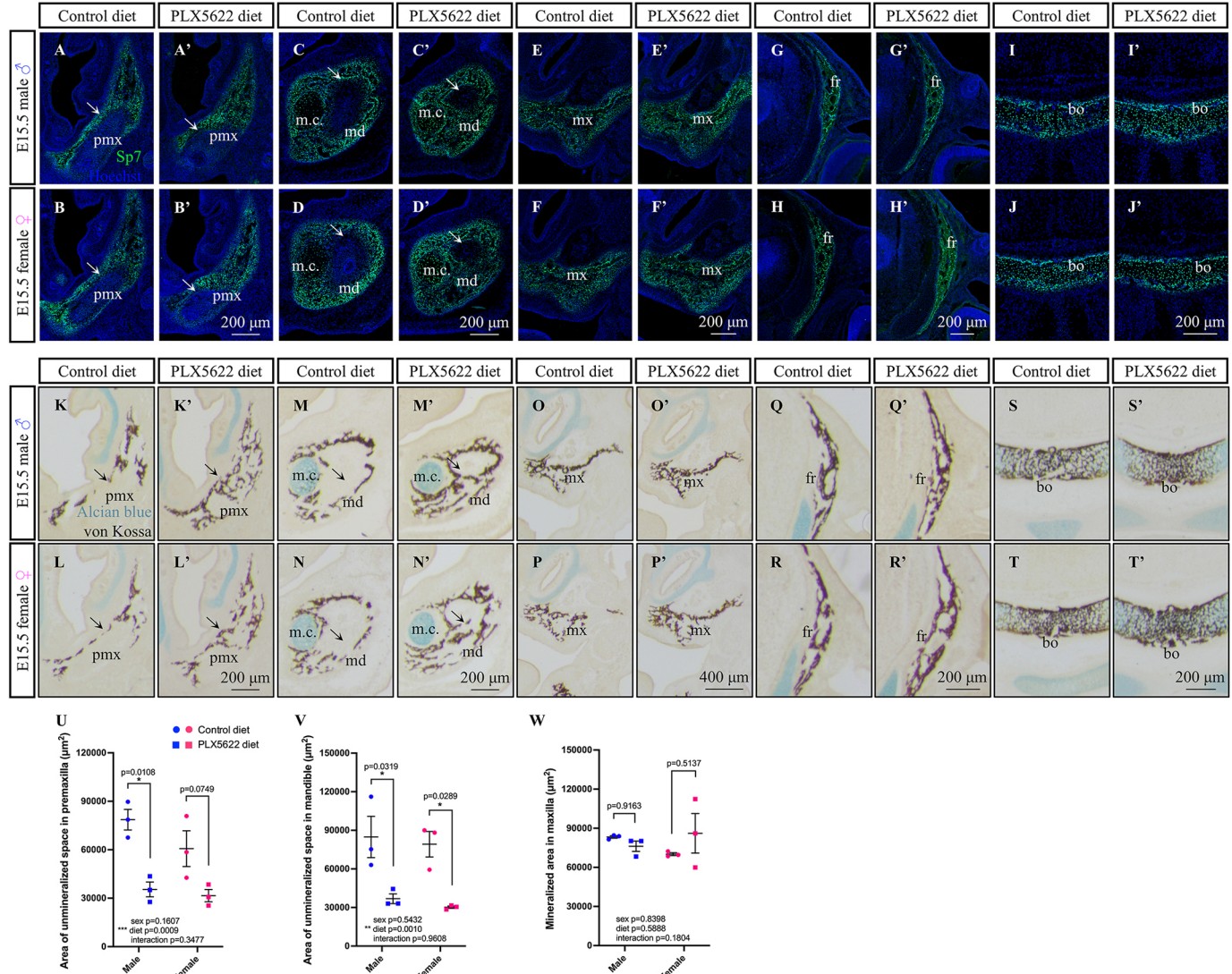

**Fig. 3. Exposure to PLX5622 during embryogenesis increases bone density in the premaxilla and mandible.** (A-J′) E15.5 Sp7 osteoblast staining in the premaxilla (A-B′) mandible (C-D′), maxilla (E-F′), frontal (G-H′) and basioccipital (I-J′) bones of CD1 embryos. (K-T′) E15.5 von Kossa and Alcian Blue staining in the premaxilla (K-L′), mandible (M-N′), maxilla (O-P′), frontal (Q-R′) and basioccipital (S-T′) bones of CD1 embryos. (U-W) Quantification of unmineralized area within the premaxilla (U) and mandible (V), and mineralized area in the maxilla (W). *n*=3 embryos per sex/treatment from two or three dams. bo, basioccipital; fr, frontal; m.c, Meckel's cartilage; md, mandible; mx, maxilla; pmx, premaxilla. Measurements represent mean±s.e.m. and were analyzed by a two-way ANOVA with Tukey's post-hoc test.

and female embryos were stained for alkaline phosphatase activity as a readout for bone mineralization. Relative to total micromass area, which was stained with Hematoxylin, control and PLX5622 frontonasal mass- and mandible-derived micromass cultures did not show a significant difference in alkaline phosphatase staining (Fig. S4A-L). Combined, these data demonstrate that prenatal exposure to PLX5622 drives region-specific disruptions to the development of the premaxilla and mandible between E12.5 and E15.5.

### Embryonic exposure to the CSF1R inhibitor PLX5622 drives sex- and strain-dependent disruptions to craniofacial bone morphogenesis

Given our findings suggest a possible increase in osteoblast differentiation and/or osteogenic potential in the premaxilla and mandible of PLX5622 embryos between E12.5 and E15.5, we were interested in whether other cartilage and/or bony phenotypes would arise late in embryogenesis. Accordingly, to visualize any phenotypes that develop late in embryogenesis, we stained P1 CD1 and C57BL/6 male and female skulls with Alcian Blue and Alizarin Red to label cartilage and bone, respectively. Consistent with previous P21 and P28 findings (Rosin et al., 2018), doming of the cranial vault was observed in all P1 PLX5622 CD1 male and female pups, with overt changes to calvarial bone size and density, including impairments to sagittal and coronal suture development (Fig. 4A-D′). Moreover, the length of the skull, interparietal bone and occipital bone were all found to significantly decrease in P1 PLX5622 CD1 male and female pups compared to control (Fig. 4E-G; main effect of diet: skull $F_{(1,20)}$=35.64, $P$<0.0001; male $P$=0.0023, female $P$=0.0021; interparietal bone $F_{(1,20)}$=49.10, $P$<0.0001; male $P$=0.0021, female $P$<0.0001; occipital bone $F_{(1,20)}$=44.85, $P$<0.0001; male $P$=0.0013, female $P$=0.0004). Similar, but milder, doming of the cranial vault was observed in P1 PLX5622 C57BL/6 male and female pups (Fig. S5A-B′, arrows). However, disruption to calvarial bone size and density, including impairments to sagittal and interfrontal sutures (Fig. S5C-D′), were still prominent in P1 PLX5622 C57BL/6 male and female pups compared to control. In contrast, skull and interparietal bone length were not significantly impacted by PLX5622 exposure (Fig. S5E,F), although occipital bone length did significantly decrease in P1 PLX5622 C57BL/6 male pups (Fig. S5G; main effect of diet $F_{(1,20)}$=13.22, $P$=0.0016; male $P$=0.0219, female $P$=0.2397).

When focusing on the P1 CD1 mandible (Fig. 4H-I′), we began to observe notable sex differences in mandible length (Fig. 4J; significant effect of diet $F_{(1,20)}$=17.88, $P$=0.0004 and diet×sex interaction $F_{(1,20)}$=4.717, $P$=0.0421) and height (Fig. 4K; significant effect of diet $F_{(1,20)}$=151.6, $P$<0.0001 and sex $F_{(1,20)}$=8.754, $P$=0.0078) in response to embryonic PLX5622 exposure. Indeed, only P1 PLX5622 CD1 female pups had shorter mandibles (Fig. 4J; male $P$=0.4821, female $P$=0.0011). While P1 PLX5622 CD1 mandible height significantly decreased in both sexes (Fig. 4K; male $P$<0.0001, female $P$<0.0001), a greater disruption was observed in P1 PLX5622 CD1 female pups (Fig. 4K; PLX5622 diet male versus female $P$=0.0313). In contrast, P1 control and PLX5622 C57BL/6 male and female mandible length was comparable (Fig. S5H-J), while mandible height was significantly decreased in both P1 PLX5622 C57BL/6 males and females (Fig. S5K; main effect of diet $F_{(1,20)}$=38.62, $P$<0.0001; male $P$<0.0001, female $P$=0.0233).

In assessing the P1 CD1 ear, we observed notable abnormalities to the ear ossicles, including an absence of the incus in PLX5622 male and female pups compared to controls (Fig. 4L-M′). Bone density was disrupted in the otic capsule and tympanic ring

(Fig. 4L-M′), where the tympanic ring was found to be significantly shorter in PLX5622 CD1 males and females alike (Fig. 4N; main effect of diet $F_{(1,20)}$=70.61, $P$<0.0001; male $P$=0.0004, female $P$<0.0001). Interestingly, we observed sex differences with respect to tympanic ring thickness (Fig. 4O; significant effect of diet $F_{(1,20)}$=43.62, $P$<0.0001 and sex $F_{(1,20)}$=11.42, $P$=0.0030). While the P1 PLX5622 CD1 tympanic ring was significantly thicker in both sexes (Fig. 4O; male $P$<0.0001, female $P$=0.0090), a larger disruption was observed in P1 PLX5622 CD1 male pups (Fig. 4O; PLX5622 diet male versus female $P$=0.0120). Prominent disruptions to the malleus and stapes were also observed when comparing P1 CD1 control and PLX5622 pups (Fig. 4L-M′), which included a significant decrease in both malleus (Fig. 4P; main effect of diet $F_{(1,20)}$=327.8, $P$<0.0001; male $P$<0.0001, female $P$<0.0001) and stapes (Fig. 4Q; main effect of diet $F_{(1,20)}$=1093, $P$<0.0001; male $P$<0.0001, female $P$<0.0001) length in both sexes in response to PLX5622 exposure. Interestingly, the P1 control C57BL/6 ear appeared less developed than what was observed at P1 in CD1 pups (compare Fig. 4L-M′ to Fig. S5L-M′), as the incus was absent. Similar to the incus, the otic capsule appeared underdeveloped in the P1 control C57BL/6 ear compared to what was observed in CD1 pups (compare Fig. 4L-M′ to Fig. S5L-M′), yet otic capsule mineralization was still found to be disrupted in P1 PLX5622 C57BL/6 male and female pups compared to control (Fig. S5L-M′, triangular arrowheads). Surprisingly, analysis of tympanic ring length (Fig. S5L-M′, arrowheads) showed sex differences (Fig. S5N; significant effect of diet $F_{(1,20)}$=19.06, $P$=0.0003 and diet×sex interaction $F_{(1,20)}$=4.939, $P$=0.0380), whereby P1 control C57BL/6 female tympanic ring length was found to be significantly shorter than male pups (Fig. S5N; $P$=0.0490), and PLX5622 exposure was only found to impact tympanic ring length in P1 PLX5622 C57BL/6 male pups (Fig. S5N; male $P$=0.0008, female $P$=0.4474). In contrast, tympanic ring thickness significantly increased in both P1 PLX5622 C57BL/6 males and females (Fig. S5O; main effect of diet $F_{(1,20)}$=97.98, $P$<0.0001; male $P$<0.0001, female $P$<0.0001). Underdevelopment of the malleus and stapes was also observed when comparing P1 C57BL/6 control and PLX5622 pups (Fig. S5L-M′), which appeared as a significant decrease in both malleus (Fig. S5P; main effect of diet $F_{(1,20)}$=27.87, $P$<0.0001) and stapes (Fig. S5Q; main effect of diet $F_{(1,20)}$=59.96, $P$<0.0001) length. While a significant decrease in malleus length was only observed in P1 C57BL/6 males (Fig. S5P; male $P$=0.0008, female $P$=0.0505), stapes length significantly decreased in both sexes (Fig. S5Q; male $P$<0.0001, female $P$=0.0029).

Intriguingly, P1 PLX5622 CD1 male and female pups also presented with disruptions to the palate, palatine sutures, and premaxilla compared to controls (Fig. 4R-S″); however, the palatine processes were only distinctly separated and sutures completely open in 50% of PLX5622 pups (Fig. 4R′,S′), compared to a milder phenotype seen in the remaining pups in which the palatine processes were in contact but smaller (Fig. 4R″,S″). Considering that the palate phenotype was severe in some pups and reminiscent of cleft palate (Funato et al., 2015; Iwata et al., 2012; Sanford et al., 1997; Zhao et al., 1999), we examined unstained P1 PLX5622 CD1 palates to confirm that no overt clefts were present (Fig. 4T-U′). Importantly, quantification of premaxilla thickness in P1 CD1 pups demonstrated that embryonic exposure to PLX5622 significantly decreased the size of the premaxilla in both sexes (Fig. 4V; main effect of diet $F_{(1,20)}$=173.8, $P$<0.0001; male $P$<0.0001, female $P$<0.0001). Abnormalities in P1 PLX5622 CD1 male and female pup cranial base development, including wider spheno-occipital and intersphenoid synchondroses, and morphological alterations to the palatine-basisphenoid sutures, could also be observed (Fig. 4W-X′).

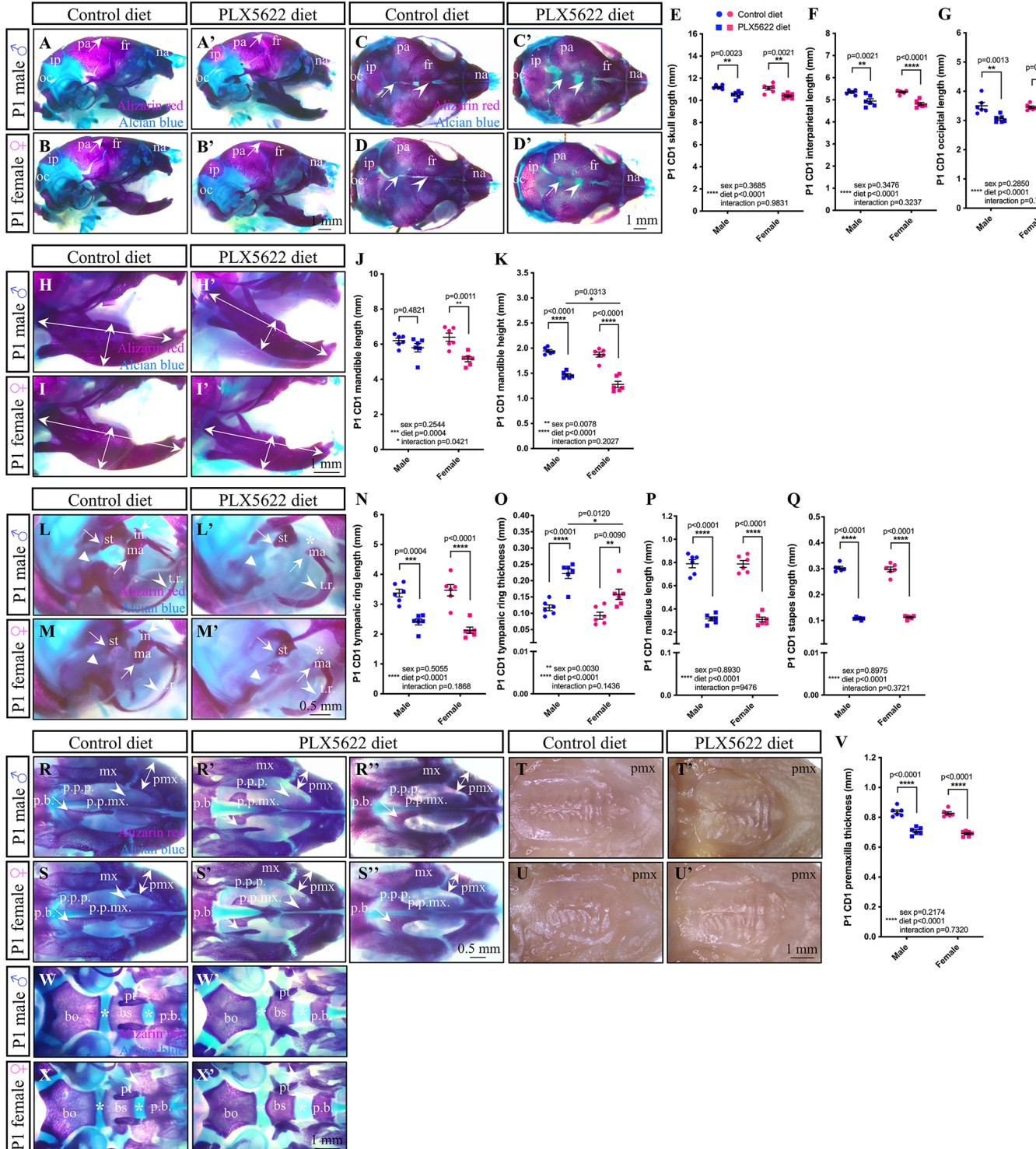

**Fig. 4. Prenatal exposure to PLX5622 disrupts craniofacial bone morphogenesis.** (A-D′) Lateral (A-B′) and dorsal (C-D′) views of P1 CD1 skulls. Arrows in A-B′ indicate skull doming. Arrowheads in C-D′ mark interfrontal suture. Arrows in C-D′ mark sagittal suture. (E-G) Quantification of skull (E), interparietal (F) and occipital (G) bone lengths. (H-K) Lateral view of P1 CD1 mandibles (H-I′) and quantification of mandible length (J) and height (K). Arrows in H-I′ indicate measured lengths. (L-Q) Lateral view of the P1 CD1 ear (L-M′) and quantification of tympanic ring length (N) and thickness (O), and malleus (P) and stapes (Q) lengths. Arrows mark incus (absent in PLX5622 pups, asterisks), stapes and malleus. Arrowheads in L-M′ mark tympanic ring. Triangular arrowheads in L-M′ mark otic capsule. (R-U′) Ventral views of the P1 CD1 palate. Arrows in R-S″ mark palatine sutures. Arrowheads in R-S″ indicate altered palate morphology. (V) Quantification of premaxilla thickness. Double-headed arrows in R-S″ indicate measured thickness. (W-X′) Ventral view of the P1 CD1 cranial base. *n*=5-8 pups per sex/treatment from two to four dams. bo, basioccipital; bs, basisphenoid; fr, frontal bone; in, incus; ip, interparietal; ma, malleus; mx, maxilla; na, nasal bone; oc, occipital; pa, parietal bone; p.b., palatine bone; pmx, premaxilla; p.p.mx., palatine process of the maxilla; p.p.p., palatine process of the palatine; pt, pterygoid process; st, stapes; t.r., tympanic ring. Measurements represent mean±s.e.m. and were analyzed by a two-way ANOVA with Tukey's post-hoc test.

Disruptions to the palate, palatine processes, and premaxilla appeared milder in Pl PLX5622 C57BL/6 male and female pups (Fig. S5R-S′); however, quantification of premaxilla thickness in P1 C57BL/6 pups demonstrated that embryonic exposure to PLX5622 significantly decreased the size of the premaxilla in both sexes (Fig. S5T; main effect of diet $F_{(1,20)}$=371.9, $P<0.0001$; male $P<0.0001$, female $P<0.0001$). Similarly, subtle abnormalities were observed in the cranial base of Pl PLX5622 C57BL/6 male and female pups, including wider spheno-occipital and intersphenoid synchondroses, while the palatine bones and palatine sutures did not appear to be impacted (Fig. S5U-V′). Taken together, these data suggest that CSF1R+ cells play important and widespread roles in craniofacial bone morphogenesis. Notably, we also observed sex differences, with male-specific phenotypes in the ear and female-specific phenotypes in the mandible, in addition to strain-dependent effects, with C57BL/6 pups presenting milder phenotypes overall.

## Skull doming in PLX5622 pups is not caused by craniosynostosis

As doming of the cranial vault is often associated with midface hypoplasia and a shortened cranial base (Vora, 2017; Eswarakumar et al., 2004; Kawasaki et al., 2017; Laurita et al., 2011; Nagata et al., 2011; Panda et al., 2013), the cranial base of P1 control and PLX5622 CD1 male and female pups were also evaluated by micro-computed tomography (μCT; Fig. 5A-D). No disruptions to posterior cranial base length or anterior frontal complex length were observed between P1 control and PLX5622 CD1 male and female pups (Fig. 5E,F). Similarly, no significant differences were observed in the lengths of the basioccipital or basisphenoid bones when comparing P1 control and PLX5622 CD1 male and female pups (Fig. 5G,H); however, the presphenoid bone was significantly shorter in P1 PLX5622 CD1 male and female pups (Fig. 5I; significant effect of diet $F_{(1,8)}$=65.07, $P<0.0001$; male $P=0.0003$, female $P=0.0203$). Although there was a significant impact of PLX5622 on the size of the spheno-occipital synchondrosis (Fig. 5J; $F_{(1,8)}$=5.740, $P=0.0435$), assessment of P1 control and PLX5622 CD1 male and female pups did not show significance (Fig. 5J; male $P=0.3106$, female $P=0.4719$). In contrast, P1 PLX5622 CD1 male and female pups presented with significantly wider inter-sphenoid synchondroses (Fig. 5K; main effect of diet $F_{(1,8)}$=31.50, $P=0.0005$; male $P=0.0066$, female $P=0.0487$).

As hydrocephalus could also be contributing to skull doming (Schmidt and Ondreka, 2019; Cohen et al., 1999; Kohn et al., 1981; Stottmann et al., 2011; Jimenez et al., 2001; Yang et al., 2019a), contrast enhanced μCT was used to examine the gross morphology of P1 control and PLX5622 CD1 male and female pup brains (Fig. 5L-S). Ventriculomegaly of the lateral and third ventricles, in addition to bilateral cavitary lesions at the cortico-striato-amygdalar boundary, could be observed both rostrally (Fig. 5L-O, arrows) and caudally (Fig. 5P-S, arrows) when comparing P1 PLX5622 CD1 male and female pups to controls. Overall, while our data suggests that PLX5622 pups do not display craniosynostosis, exposure to PLX5622 during embryogenesis does drive disruptions to the cranial base and structural changes to the gross morphology of the brain.

## Gestational exposure to PLX5622 alters cytokine and chemokine secretion from CSF1R+ cells

Given that macrophages are known to secrete cytokines and chemokines, which are capable of stimulating tissue growth postnatally (Keshvari et al., 2021; Gow et al., 2010) and during wound healing (Ploeger et al., 2013; Weber et al., 1999), we next investigated whether PLX5622 exposure disrupts CSF1R+ cell-derived secreted factors to contribute to the reported craniofacial phenotypes. Luminex analysis of cytokines and chemokines secreted from cultured E13.5 control and PLX5622 C57BL/6 craniofacial tissues (Fig. 6A) showed a significant downregulation of CCL2, CCL3, CCL4, CCL5, CCL12, CCL22, CXCL1, CXCL2, CXCL10, G-CSF (CSF3), IFNβ1, IL6 and TNFα (Fig. 6B-G,I-O; main effect of diet: CCL2 $F_{(1,32)}$=113.7, $P<0.0001$; CCL3 $F_{(1,32)}$=68.72, $P<0.0001$; CCL4 $F_{(1,32)}$=90.93, $P<0.0001$; CCL5 $F_{(1,32)}$=18.84, $P=0.0001$; CCL12 $F_{(1,32)}$=120.1, $P<0.0001$; CCL22 $F_{(1,32)}$=46.38, $P<0.0001$; CXCL1 $F_{(1,32)}$=20.25, $P<0.0001$; CXCL2 $F_{(1,32)}$=117.2, $P<0.0001$; CXCL10 $F_{(1,32)}$=59.47, $P<0.0001$; G-CSF $F_{(1,32)}$=70.92, $P<0.0001$; IFNβ1 $F_{(1,32)}$=15.89, $P=0.0004$; IL6 $F_{(1,32)}$=71.26, $P<0.0001$; TNFα $F_{(1,32)}$=137.4, $P<0.0001$) and a significant upregulation of CX3CL1 (Fig. 6H; main effect of diet $F_{(1,32)}$=30.35, $P<0.0001$) upon PLX5622 exposure. Interestingly, CCL2 (Fig. 6B; male $P<0.0001$, female $P<0.0001$), CCL3 (Fig. 6C; male $P<0.0001$, female $P<0.0001$), CCL4 (Fig. 6D; male $P<0.0001$, female $P<0.0001$), CCL12 (Fig. 6F; male $P<0.0001$, female $P<0.0001$), CCL22 (Fig. 6G; male $P=0.0001$, female $P=0.0003$), CXCL2 (Fig. 6J; male $P<0.0001$, female $P<0.0001$), CXCL10 (Fig. 6K; male $P<0.0001$, female $P<0.0001$), G-CSF (Fig. 6L; male $P=0.0001$, female $P<0.0001$), IL6 (Fig. 6N; male $P=0.0003$, female $P<0.0001$) and TNFα (Fig. 6O; male $P<0.0001$, female $P<0.0001$) were downregulated in both E13.5 PLX5622 male and female craniofacial tissue cultures, while CCL5 (Fig. 6E; male $P=0.2950$, female $P=0.0007$), CXCL1 (Fig. 6I; male $P=0.0838$, female $P=0.0025$) and IFNβ1 (Fig. 6M; male $P=0.2697$, female $P=0.0034$) were only found to be significantly downregulated in E13.5 PLX5622 female craniofacial tissue cultures. In contrast, CX3CL1 was significantly upregulated in both sexes (Fig. 6H; male $P=0.0002$, female $P=0.0255$). Although 34 other cytokines and chemokines were assessed in our analyses, these secreted factors were either lowly expressed, not significantly changed, or below the level of detection (Table S3).

To confirm that these cytokines and chemokines could indeed be derived from CSF1R+ cells in craniofacial tissues, we analyzed publicly available single-cell RNA sequencing datasets from E12.5 and E13.5 FVB/NJ and C57BL/6J craniofacial mesenchyme (Angelozzi et al., 2022; Rajderkar et al., 2024). *Csf1r*-expressing cells extracted from these datasets formed three distinct clusters, with clusters 1 and 2 upregulating macrophage markers such as *Ptprc* and *Lyve1*, while cluster 3 expressed the osteoclast-specific marker *Ctsk* (Fig. S6A). Cluster 1 was found to express *Ccl2*, *Ccl3*, *Ccl4* and *Ccl12*, while cluster 2 only upregulated *Ccl12* (Fig. S6A). Other cytokines and chemokines, such as *Cxcl2*, *Cxcl10*, *Tnf*, etc., were not found to be upregulated in any of the clusters (Fig. S6A). Importantly, assessment of mRNA expression in the E15.5 mandible, maxilla and ear (Fig. S6B-D) using fluorescence *in situ* hybridization revealed the presence of *Csf1r*/*Ccl4* double-positive cells in each craniofacial tissue examined (Fig. S6E-G). Moreover, immunofluorescence staining of E15.5 craniofacial tissue cryosections also confirmed the presence of CCL3 and CCL4 in *Csf1r*EGFP+ cells in each of the craniofacial tissues assessed (Fig. S6H-S).

Given that macrophages can themselves respond to secreted signals (Biguetti et al., 2018; Namangkalakul et al., 2023; Xu et al., 2022; Leichtle et al., 2010; Repeke et al., 2010; Takahashi et al., 2002; Wang et al., 2013; Cheung et al., 2009; von Stebut et al., 2003; Huang et al., 2020; Ma et al., 2025; Bao et al., 2022; Herbold et al., 2010; Nie et al., 2024; Caetano et al., 2023; Lee et al., 2017; Tsai et al., 2019; Calvo et al., 1996; Kaplanski et al., 2003), we were

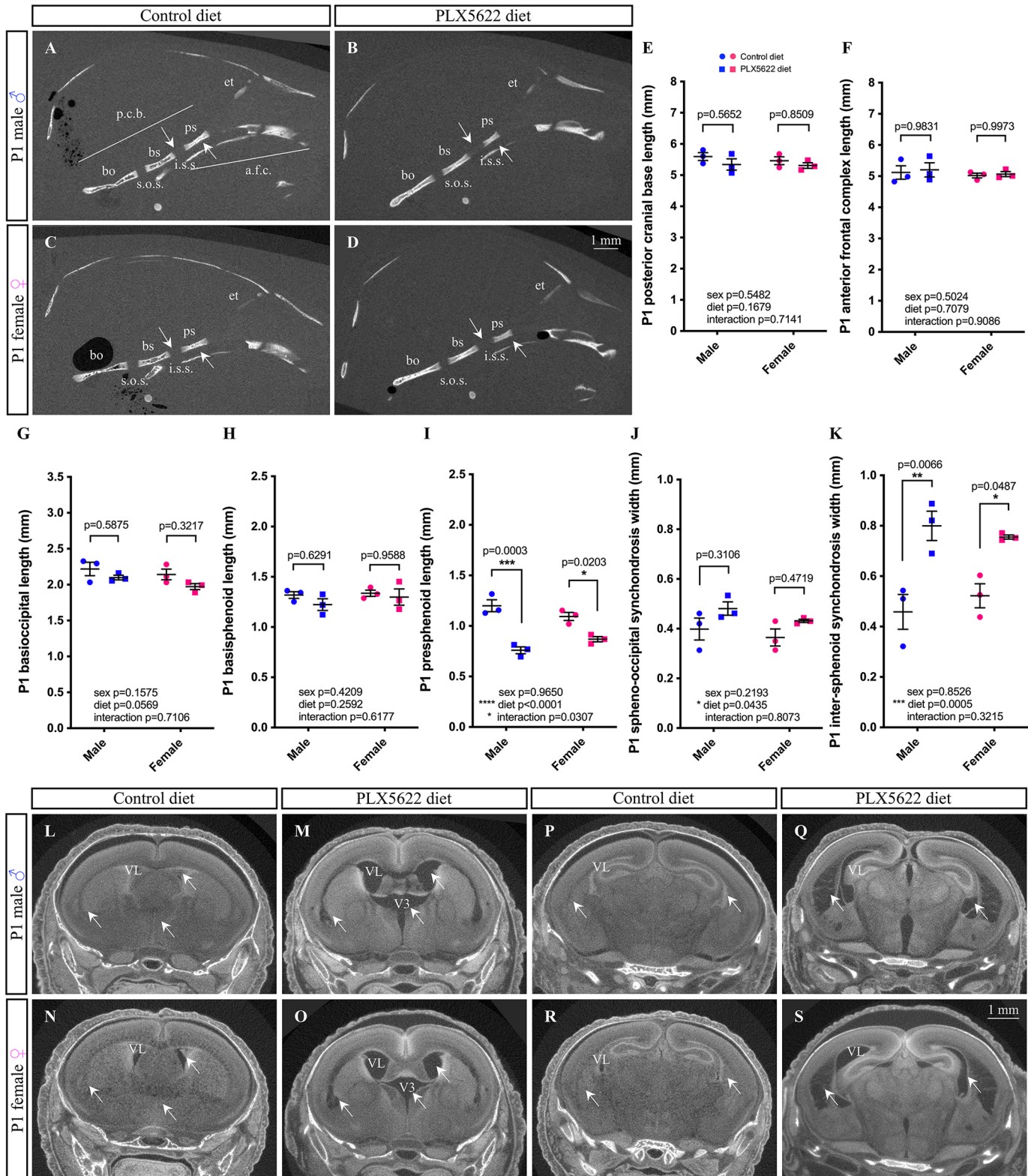

**Fig. 5. Prenatal exposure to PLX5622 disrupts cranial base and brain development.** (A-D) Lateral view of µCT scans of P1 CD1 skulls show presphenoid bone and inter-sphenoid synchondrosis. (E-K) Quantification of posterior cranial base length (E), anterior frontal complex length (F), basioccipital length (G), basisphenoid length (H), presphenoid length (I), spheno-occipital synchondrosis width (J) and inter-sphenoid synchondrosis width (K). (L-S) Coronal view of µCT scans of P1 CD1 skulls across the rostral (L-O) to caudal (P-S) axis. Arrows mark lateral (VL) and third (V3) ventricles. n=3-5 pups per sex/treatment from two or three dams. a.f.c., anterior frontal complex; bo, basioccipital; bs, basisphenoid; et, ethmoid; i.s.s., inter-sphenoid synchondrosis; p.c.b., posterior cranial base; ps, presphenoid bone; s.o.s., spheno-occipital synchondrosis. Measurements represent mean±s.e.m. and were analyzed by a two-way ANOVA with Tukey's post-hoc test.

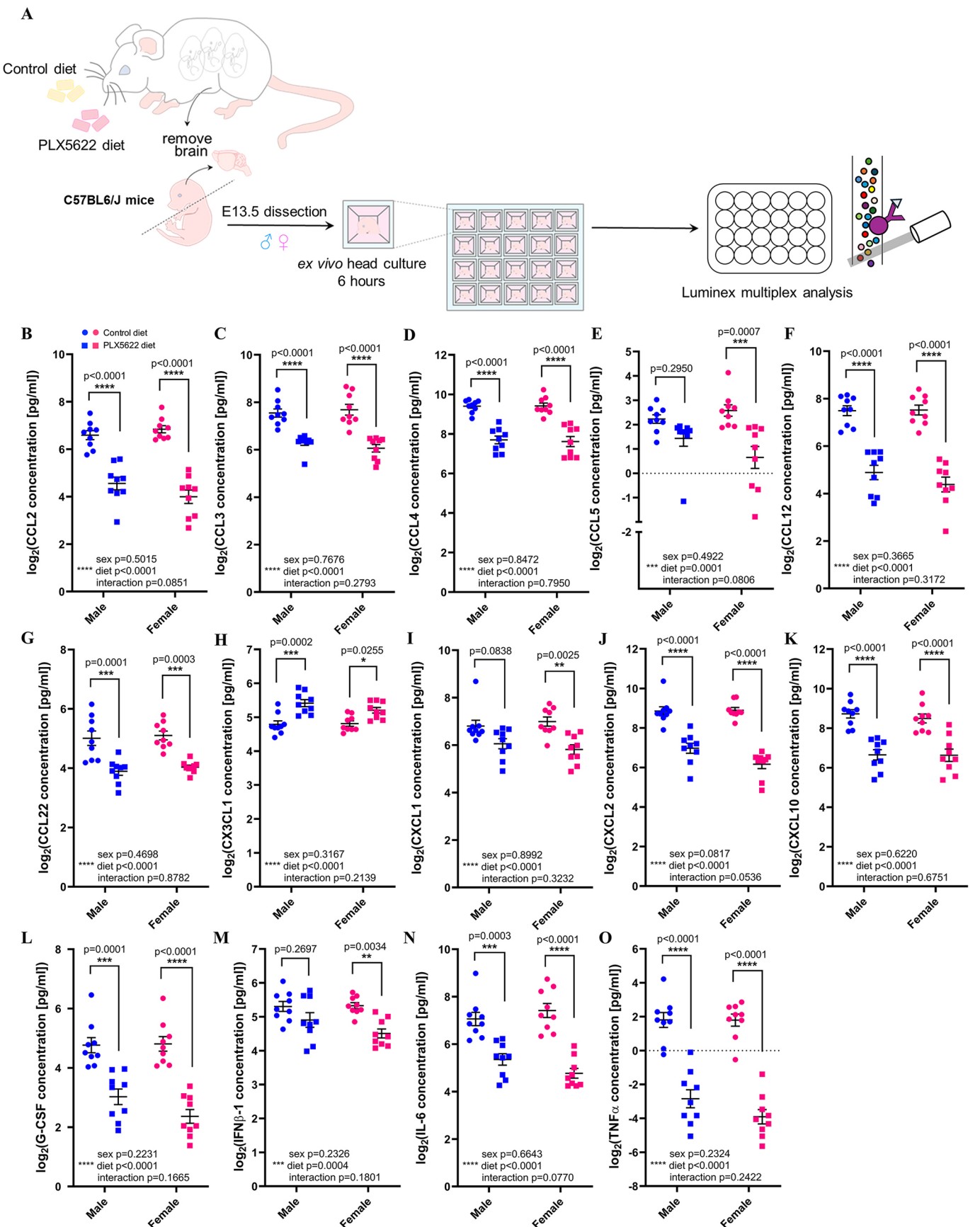

**Fig. 6. Exposure to PLX5622 during embryogenesis disrupts cytokine and chemokine secretion in craniofacial tissue cultures.** (A) Schematic illustrating craniofacial tissue culture and Luminex multiplex analysis. (B-O) Cytokine/chemokine analysis of E13.5 craniofacial tissue culture media. *n*=9 per sex/treatment from three dams. Quantifications represent mean±s.e.m. and were analyzed by a two-way ANOVA with Tukey's post-hoc test.

interested in whether treatment with the cytokines and chemokines found to be altered would impact macrophage physiology, such as phagocytosis (Fig. S7A). Although incubation of E13.5 craniofacial tissue with a mixture of CCL2, CCL3, CCL4, CCL12, CXCL2, CXCL10 and IL6 did not increase phagocytosis of pHrodo BioParticles by $Csf1r^{EGFP+}$ cells ex vivo (Fig. S7B), we observed interactions between $Csf1r^{EGFP+}$ macrophages and $Wnt1^{Cre}$-driven tdTomato$^+$ NCCs across numerous embryonic craniofacial structures such as the premaxilla and mandible (Fig. S7C-J), which has been observed in human embryonic skin (Wang et al., 2023). Combined, these data suggest that embryonic exposure to PLX5622 significantly disrupts cytokine and chemokine levels in craniofacial tissues, with these changes likely being primarily driven by CSF1R$^+$ cells and suggest that CSF1R$^+$ cells may signal to and physically interact with NCCs to impact NCC developmental programs.

### CSF1R$^+$ cell depletion disrupts neural crest proliferation

To test whether the depletion of CSF1R$^+$ cells and altered cytokine/chemokine signaling impact surrounding cells to influence craniofacial morphogenesis, we focused on NCCs as many of the craniofacial structures disrupted by PLX5622 exposure are derived from this lineage. To start, we employed a neural crest-derived sphere assay to examine the impact of PLX5622 exposure on NCCs (Fig. 7A). Briefly, neural crest-derived cells were isolated from E12.5 $Wnt1^{Cre}$;$Rosa26^{tdTomato}$ control and PLX5622 male and female embryos and cultured in 24-well plates for 10 days to allow for sphere formation (Lewis et al., 2013). Primary spheres were then dissociated, and cells were cultured for an additional 10 days to form secondary spheres. Primary and secondary neural crest-derived spheres retained $Wnt1^{Cre}$-driven tdTomato$^+$ signal and also expressed NCC markers such as PDGFRα (Schatteman et al., 1992), SOX2 (Cai et al., 2002; Roellig et al., 2017) and SOX10 (Kim et al., 2003; Southard-Smith et al., 1998), regardless of whether spheres were derived from E12.5 control or PLX5622 embryos (Fig. 7B-M). Intriguingly, gestational exposure to PLX5622 resulted in a significant decrease in the number of primary neural crest-derived spheres in both sexes (Fig. 7N; significant effect of diet $F_{(1,25)}=51.00$, $P<0.0001$; male $P=0.0098$, female $P<0.0001$), suggesting a reduced proliferative capacity of NCCs.

To validate that embryonic NCC proliferation was indeed decreased in response to gestational PLX5622 exposure, we performed immunofluorescence staining on E15.5 $Wnt1^{Cre}$;$Rosa26^{tdTomato}$ craniofacial tissue cryosections to quantify tdTomato$^+$ cell numbers and proliferation using a Ki67 antibody. $Wnt1^{Cre}$-driven tdTomato$^+$ signal was found to be highly expressed within mineralizing regions of craniofacial bones, particularly in regions where we had previously identified abnormal embryonic osteoblast staining/mineralization in response to PLX5622 exposure. Accordingly, we focused our analyses on the E15.5 premaxilla and mandible. Quantification of $Wnt1^{Cre}$-driven tdTomato$^+$ and Ki67$^+$ cells (Fig. 7O-T, arrows) in the E15.5 male and female premaxilla showed a significant decrease in both tdTomato$^+$ neural crest-derived cells (Fig. 7Q; main effect of diet $F_{(1,8)}=50.24$, $P=0.0001$; male $P=0.0043$, female $P=0.0049$) and Ki67$^+$ proliferating cells (Fig. 7T; main effect of diet $F_{(1,8)}=74.75$, $P<0.0001$; male $P=0.0009$, female $P=0.0018$) in PLX5622 embryos compared to control. Similarly, quantification of $Wnt1^{Cre}$-driven tdTomato$^+$ and Ki67$^+$ cells (Fig. 7U-Z, arrows) in the E15.5 male and female mandible showed a significant decrease in tdTomato$^+$ cells (Fig. 7W; significant effect of diet $F_{(1,8)}=68.58$, $P<0.0001$; male $P=0.0003$, female $P=0.0118$) and Ki67$^+$ cells (Fig. 7Z; main effect of diet $F_{(1,8)}=68.25$, $P<0.0001$; male

$P=0.0007$, female $P=0.0046$) in PLX5622 embryos compared to control. Interestingly, E15.5 male embryos showed a more severe decrease in tdTomato$^+$ cells in the mandible in response to PLX5622 exposure (Fig. 7W; significant effect of sex $F_{(1,8)}=14.64$, $P=0.0050$; PLX5622 male versus female $P=0.0111$). Taken together, these data suggest that the proliferative capacity of cranial NCCs decreases in response to PLX5622 exposure.

## DISCUSSION

This study is the first to provide an in-depth characterization of CSF1R$^+$ macrophage and osteoclast depletion in craniofacial tissues across embryogenesis in response to PLX5622 exposure (Rosin et al., 2018; Nagra et al., 2023; Yongzhen et al., 2024; Bosch et al., 2023; Elmore et al., 2018; Hennen et al., 2025; Lei et al., 2020; Spangenberg et al., 2019), a pharmacological tool that allows for temporal control and transient depletion of CSF1R-expressing cells. Exposure to PLX5622 significantly depleted CSF1R$^+$ cells across embryogenesis, and either halts or significantly delays osteoclastogenesis, leading to a loss of bone resorptive activity throughout the developing craniofacial bones in the embryo. To our surprise, depleting CSF1R+ cells using PLX5622 does not impact the gross morphology of craniofacial nerves or muscles in male or female CD1 or C57BL/6 embryos. In contrast, craniofacial defects in the premaxilla and mandible appear as early as E15.5, the earliest stage at which a craniofacial phenotype has been reported in a CSF1R-disrupted model, while abnormalities in skull shape, cranial sutures, ear ossicles, palate and cranial base development appear to arise later in embryogenesis, between E15.5 and birth. Although our data suggest that craniosynostosis does not account for the skull doming in P1 PLX5622 pups, disruptions to the cranial base and/or structural changes to the gross morphology of the brain could impact skull development and contribute to skull doming. Notably, we observed sex differences in P1 PLX5622 pups, whereby males presented with more prominent ear phenotypes and females displayed greater disruptions to the mandible. We also observed strain-dependent effects, with overall stronger phenotypes being observed in CD1 pups than in C57BL/6 pups. Interestingly, embryonic exposure to PLX5622 also significantly altered cytokine and chemokine levels in craniofacial tissues, with these changes likely being primarily driven by CSF1R$^+$ cells, namely macrophages. These changes in signaling within craniofacial tissues have the potential to alter CSF1R$^+$ cell dynamics and interactions with neighboring cells, which could explain the observed decrease in proliferative capacity of NCCs in response to PLX5622 exposure. Altogether, a reduction in NCC proliferation in craniofacial tissues could contribute to alterations in differentiation and osteogenic potential, resulting in the phenotypic disruptions in the premaxilla and mandible, and likely the many other bony abnormalities seen in PLX5622 offspring.

Here, we used both a $Csf1r^{EGFP}$ transgenic fluorescent reporter and immunostaining against CSF1R to characterize CSF1R$^+$ cell depletion in response to PLX5622 exposure. Although we found EGFP signal in cells negative for CSF1R immunostaining, this could be due to variable timing between $Csf1r^{EGFP}$ transgene expression and CSF1R protein or, alternatively, EGFP signal could be retained (de Luis et al., 2025; Snapp, 2005; Stadler et al., 2013). In response to PLX5622 exposure from E3.5 onwards, CSF1R$^+$ cell depletion was observed in all tissues examined and remained relatively stable (~50%) across embryogenesis, albeit with notable sex differences in the timing of depletion. Apoptotic cells accumulated predominantly in nervous and muscular structures and not in or around developing cartilaginous/bony tissues, except

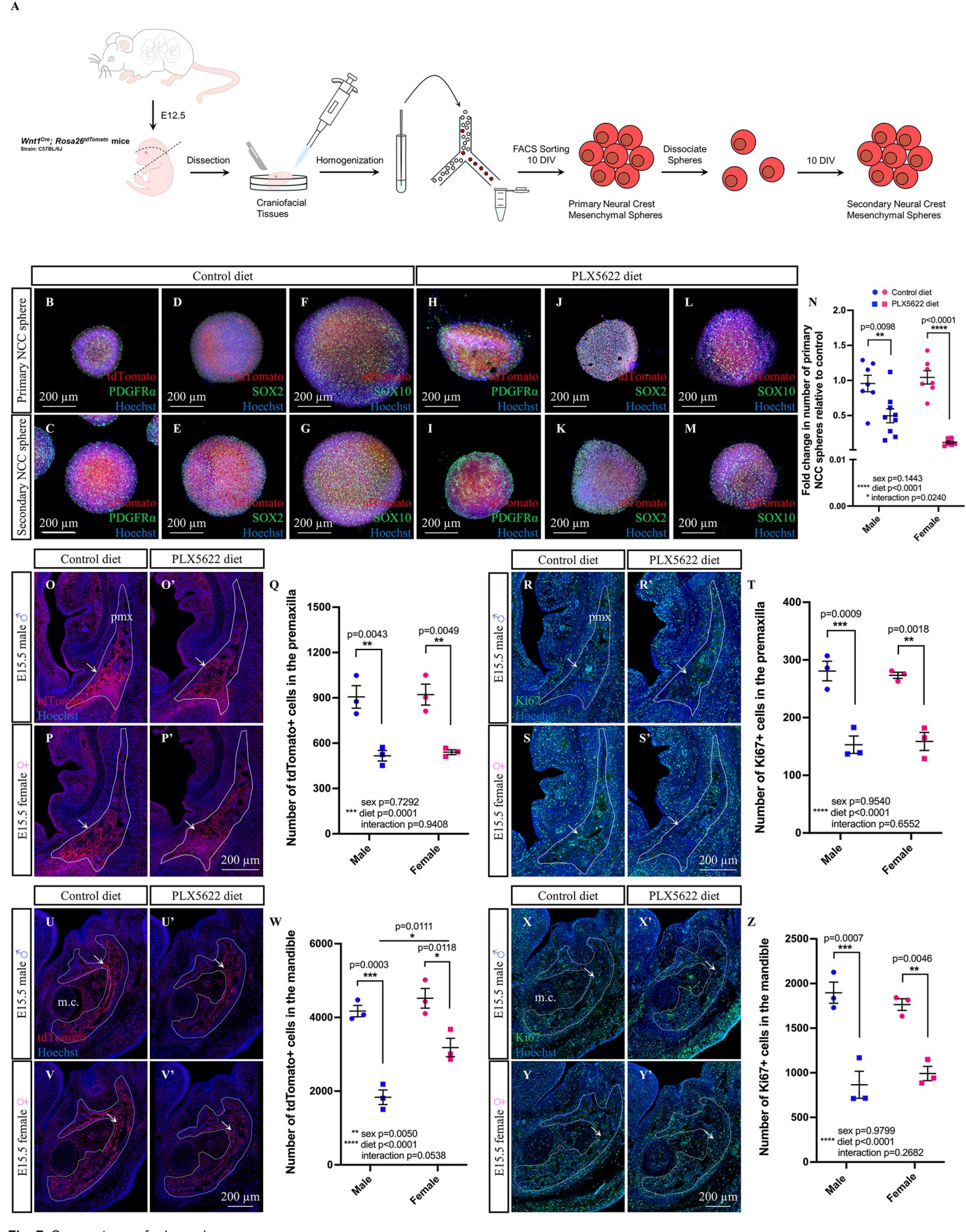

**Fig. 7.** See next page for legend.

**Fig. 7. Gestational exposure to PLX5622 decreases neural crest proliferation.** (A) Schematic illustrating sphere assay. (B-M) E12.5 *Wnt1^Cre^; Rosa26^tdTomato^* neural crest-derived spheres express PDGFRα (B,C,H,I), SOX2 (D,E,J,K) and SOX10 (F,G,L,M). (N) Primary sphere quantification (*n*=6-9 embryos per sex/treatment from two or three dams). (O-Z) Immunofluorescence images of E15.5 *Wnt1^Cre^*-driven tdTomato⁺ signal (O-P′,U-V′) and Ki67 staining (R-S′,X-Y′) in the premaxilla (O-T) and mandible (U-Z). Quantification of tdTomato⁺ (Q,W) and Ki67⁺ (T,Z) cells. White lines mark the tissue regions used for quantifications (*n*=3 embryos per sex/treatment from three dams). DIV, days *in vitro*; m.c., Meckel's cartilage; pmx, premaxilla. Counts represent mean±s.e.m. and were analyzed by a two-way ANOVA with Tukey's post-hoc test.

for the ear. Indeed, programmed cell death of mesenchyme between the lens and retina at E10 in C57BL/6J mice is essential for normal eye development (Silver and Hughes, 1974), while caspase-3 deficient C57BL/6 mice present with inner ear disruptions in the cochlea and vestibule and severe auditory dysfunction, demonstrating that caspase-3-dependent apoptosis is essential for normal inner ear development and function (Makishima et al., 2011). It is interesting to note that eye defects were previously reported in some PLX5622-exposed mice, where one eye could not be opened until later in adulthood (Rosin et al., 2018). Therefore, increased apoptosis resulting from PLX5622 exposure could perturb normal apoptotic programs and disrupt craniofacial morphogenesis, resulting in nerve and/or muscle phenotypes that could appear after E15.5, especially since bone disruptions exist across nearly the entire PLX5622 head and changes in the shape of mineralized tissue can directly impact craniofacial nerve and muscle development (Wadu et al., 1997; Yamamoto et al., 2020; Shen et al., 2015; Kitai et al., 2002; Yamamoto et al., 2018; Toth et al., 2013; Iyyanar et al., 2022). For example, the shape of the temporalis muscle can change upon deformation of the mandibular condyle, and the size of neurovascular bundles can be reduced upon removal of teeth from the mandible (Wadu et al., 1997; Yamamoto et al., 2020). Accordingly, craniofacial nerve and muscle development in PLX5622-exposed offspring should be studied postnatally to confirm whether phenotypes eventually develop.

Interestingly, despite CSF1R⁺ cell deficiency, *Csf1r* KO rat phenotypes appear to develop postnatally, with normal placental and embryonic development reported (Hume et al., 2023); however, there is a paucity of literature focusing on CSF1R⁺ cell-deficient craniofacial phenotypes during the embryonic period (see Table S4). Indeed, *Csf1/Csf1r*-disrupted genetic mouse models show craniofacial phenotypes consistently at P10, but as early as P6 (Dai et al., 2002, 2004; Marks and Lane, 1976; Okano and Kishimoto, 2019; Harris et al., 2012; Ryan et al., 2001). Here, our pharmacological CSF1R⁺ cell depletion model displays embryonic bone disruptions as early as E15.5 and multiple craniofacial bone abnormalities at P1, the vast majority of which are consistent with *Csf1/Csf1r*-disrupted genetic models (Table S4). As *Csf1/Csf1r*-disrupted genetic models and the PLX5622 pharmacological depletion model similarly display domed skulls (Dai et al., 2002, 2004; Marks and Lane, 1976; Pridans et al., 2018; Okano and Kishimoto, 2019; Rosin et al., 2018; Nagra et al., 2023; Harris et al., 2012; Ryan et al., 2001) (Table S4), and skull doming can be caused by midface hypoplasia and inhibited cranial base expansion (Vora, 2017; Eswarakumar et al., 2004; Kawasaki et al., 2017; Laurita et al., 2011; Nagata et al., 2011; Panda et al., 2013), we thoroughly assessed the cranial base. Although cranial base length was unaffected by PLX5622 exposure, suggesting that craniosynostosis and/or cranial base hypoplasia is likely not responsible for the skull doming seen in PLX5622 pups, the smaller cranial base bones and widened synchondroses observed in PLX5622

mice could contribute to the skull phenotype. We also noted enlarged lateral and third ventricles in PLX5622 pups, suggesting that mild hydrocephalus could contribute to skull doming, which has been seen in other animal models (Schmidt and Ondreka, 2019; Cohen et al., 1999; Kohn et al., 1981; Stottmann et al., 2011; Jimenez et al., 2001; Yang et al., 2019a). Moreover, we observed brain lesions at the cortico-striato-amygdalar boundary in PLX5622 pups, an area susceptible to physiological morphogenetic stress that is essential for fetal brain morphogenesis (Lawrence et al., 2024). Given that microglia play crucial roles in maintaining structural integrity at this boundary (Lawrence et al., 2024), the loss of microglia upon PLX5622 exposure (Rosin et al., 2018) could disrupt normal maintenance and repair mechanisms, resulting in mechanical pressures on the cranial vault and skull doming; however, skull doming has not been reported in *Csf1r^ΔFIRE/ΔFIRE^* mice, which show similar brain lesions and loss of structural integrity (Lawrence et al., 2024; Rojo et al., 2019) (Table S4). Therefore, it is also possible that insufficient mineralization compromises the integrity of the calvaria such that it bulges outward in response to normal mechanical forces of the developing brain. Interestingly, the morphological disruptions observed in the brain and cranial base of PLX5622 mice are reminiscent of the brain abnormalities and cranial base sclerosis observed in individuals with BANDDOS (brain abnormalities, neurodegeneration and dysosteosclerosis), a human disorder caused by bi-allelic variants in *CSF1R* (Guo et al., 2019; Monies et al., 2017; Beerepoot et al., 2024; Jiang et al., 2022; Kindis et al., 2021; Oosterhof et al., 2019, 2018; Tamhankar et al., 2020; Daghagh et al., 2023). The skull doming, cranial suture impairments and cranial base disruptions seen in PLX5622 mice could be a useful model for studying the cranial vault malformations observed in BANDDOS (Guo et al., 2019; Monies et al., 2017; Beerepoot et al., 2024; Kindis et al., 2021), especially considering the titratable nature of PLX5622, as CSF1R inhibition could more accurately reflect hypomorphic *CSF1R* variants, which account for 8/11 identified variants associated with BANDDOS (Dulski et al., 2023).

Although the phenotypes seen in *Csf1r/CSF1R*-disrupted rodents and humans are of an osteosclerotic nature (Dai et al., 2002, 2004; Okano and Kishimoto, 2019; Keshvari et al., 2021; Guo et al., 2019; Monies et al., 2017; Beerepoot et al., 2024; Kindis et al., 2021; Inoue et al., 2023; Nandi et al., 2006), likely due to a lack of osteoclasts, we observed decreased bone formation in the calvaria of PLX5622 pups and disruptions in the development of the premaxilla, mandible, ear bones, palate and cranial base (Table S4). These phenotypes are unlikely to result solely from a loss of bone resorption by osteoclasts. Instead, prenatal exposure to PLX5622 could disrupt signaling from CSF1R⁺ macrophages that are required for bone formation and growth (Keshvari et al., 2021; Batoon et al., 2024a,b, 2019; Alexander et al., 2011; Blumer et al., 2008; Chang et al., 2008). Intriguingly, these osteogenic pathways may also respond to PLX5622 differently between sexes given the sex-specific mandible and ear phenotypes observed and sex-dependent changes in several cytokines/chemokines. CSF1R⁺ osteoclasts are also more abundant and active in long bones of female mice compared to males, resulting in decreased bone mass (Mun et al., 2021; Herbert et al., 2015; Paglia et al., 2016), and ovariectomy-induced estrogen deficiency increases CSF1 production (Cenci et al., 2000; Kimble et al., 1996; Srivastava et al., 1998), which together further suggest a potential for sex hormone-dependent responses to CSF1R⁺ cell depletion following PLX5622 exposure. It is also interesting to note that we observed basal sex differences that were independent of PLX5622 exposure, which are in line with sex differences observed in craniofacial bones that can be observed

at birth in humans (Przystanska et al., 2020), and highlight the importance of studying both sexes in craniofacial research.

Interestingly, the impact of PLX5622 exposure on osteogenesis appears to be independent of whether the bone is formed by intramembranous (calvaria) or endochondral ossification (auditory ossicles) (Hall, 1988; Marvaso and Bernard, 1977; Sandberg, 1991; Takarada et al., 2016; Zimmermann, 1992; Hall and Miyake, 2000; Mallo, 1998; Miyake et al., 1996; Wood et al., 2010). This suggests that signaling alterations may act on a common progenitor, which is supported by our findings as PLX5622 exposure alters cytokine/chemokine levels and NCCs give rise to the majority of the craniofacial bones found to be disrupted in PLX5622 mice (Chai et al., 2000; Jiang et al., 2002; Jaenisch, 1985; Tan and Morriss-Kay, 1986; Yoshida et al., 2008; Bildsoe et al., 2013). Indeed, NCCs do respond to several of the cytokines/chemokines disrupted by PLX5622, including CCL2, which induces NCC migration into the mouse heart (Tamura et al., 2011), and CXCL1 and IL6, which mediate NCC differentiation into osteoblasts during intramembranous ossification in a mouse calvarial defect model (Kamalakar et al., 2021). CCL2 and CXCL1 are also chemoattractants for osteoclast precursors and osteoclastogenesis (Hu et al., 2020; Miyamoto et al., 2009; Sul et al., 2012; Tamasi et al., 2013; Hardaway et al., 2015; Onan et al., 2009), while altering the secretion of any number of the other cytokines/chemokines found to change in our study could impact osteoclast function and bone remodeling (Lee et al., 2017, 2018; Hoshino et al., 2010; Taddei et al., 2013; Wintges et al., 2013; Yu et al., 2023; Araujo-Pires et al., 2015; Ha et al., 2010, 2011; Ma et al., 2023; Yang et al., 2019b; Yano et al., 2005; Kwak et al., 2008; Marahleh et al., 2019; Takahashi et al., 1996; Takayanagi et al., 2002; Yu et al., 2021; Zhang et al., 2001; Zhao et al., 2014). Uniquely, IL6 has been shown to be an important regulator of bone remodeling by negatively regulating osteoblasts and modulating osteoclast numbers and resorption (Kaneshiro et al., 2014; Kitamura et al., 1995; Liu et al., 2014; Palmqvist et al., 2002; Poli et al., 1994). The downregulation of these signals in PLX5622-exposed mice could impact recruitment and differentiation of osteoblast and osteoclast precursors and disrupt bone remodeling processes necessary for normal craniofacial bone morphogenesis. Specifically, the decreases in bone size could be caused by reduced NCC proliferation and subsequent NCC deficiency, while the increased bone density observed from E15.5 onwards may be driven by decreased bone resorption and remodeling (Hayden et al., 1995; Howard et al., 1981; Nakamura et al., 2003; Tang et al., 2009; Zhao et al., 2006). Alternatively, increased osteogenesis could reflect premature NCC differentiation into osteoblasts, possibly in response to altered CXCL1 and IL6 signaling (Kamalakar et al., 2021), resulting in NCC progenitor deficiencies (Hu et al., 2011; Ngan et al., 2011; Rabbani et al., 2011). Although it is unclear how the disruptions to NCCs and osteoclast precursors/osteoclastogenesis culminate in the phenotypic disruptions seen in PLX5622 offspring, our findings suggest that CSF1R[+] macrophages are a likely source for the many cytokines/chemokines found to be altered in response to PLX5622 exposure and may mediate NCC and/or osteoclast deficiencies.

Herein, we demonstrated that CSF1R[+] cells are required for development of the skull, cranial sutures, premaxilla, mandible, ear ossicles, palate and cranial base. Additionally, we showed that embryonic exposure to PLX5622 drives sex-, strain- and time-dependent disruptions in craniofacial development. Together, these data highlight an underappreciated role for CSF1R[+] macrophages and osteoclasts in craniofacial morphogenesis during the embryonic period and suggest that altered interactions with and/or signaling to NCCs may underlie the phenotypes observed in PLX5622 offspring.

## Limitations of the study

A potential limitation of our study is our use of the CSF1R inhibitor PLX5622 to deplete CSF1R-expressing cells. It has been reported that PLX5622 could have off-target effects on non-CSF1R-expressing brain endothelial cells, potentially due to inhibition of the CSF1R-related tyrosine kinases FLT3, KIT, AURKC and/or KDR (Profaci et al., 2024). KIT and KDR may be expressed by embryonic NCCs (Faure et al., 2020; Hudacova et al., 2025; Luo et al., 2003; Motohashi et al., 2014, 2011; Wilson et al., 2004; Yamane et al., 1999), suggesting that PLX5622 could have off-target impacts on NCCs; therefore, future studies should examine changes to fetal NCCs in *Csf1r* KO mice. However, it is important to note that PLX5622 is more than 50-fold more active against CSF1R than KIT and KDR in cell-free enzyme assays (Spangenberg et al., 2019). The effects on endothelial cells are also specific to the central nervous system (Profaci et al., 2024). Indeed, brain alterations could impact development of the skull or surrounding craniofacial tissue, although the postnatal brain phenotypes described herein match those of *Csf1r*[ΔFIRE/ΔFIRE] and other microglia-depleted mice (Lawrence et al., 2024), and the skull doming is consistent with phenotypes in *Csf1/Csf1r*-disrupted genetic models, suggesting that these phenotypes are not driven by disruption to brain endothelial cells (Dai et al., 2002; Marks and Lane, 1976; Pridans et al., 2018; Okano and Kishimoto, 2019; Dai et al., 2004; Harris et al., 2012; Ryan et al., 2001).

## MATERIALS AND METHODS

### Experimental model details

Animal work was carried out in accordance with guidelines and regulations of the Canadian Council of Animal Care and received prior approval from the University of British Columbia's Animal Care Committee (protocols A21-0170 and A21-0171). CD1 (CR: 022; RRID: IMSR_CRL:022, Charles River) and C57BL/6 (JAX: 005304; RRID: IMSR_JAX:005304; The Jackson Laboratory) embryos and P1 pups were utilized across the study, as described. *Csf1r*[EGFP] mice (JAX: 005304; RRID: IMSR_JAX:005304; The Jackson Laboratory) were crossed to C57BL/6 mice to generate heterozygous embryos for flow cytometry and immunofluorescence experiments. *Wnt1*[Cre] mice (JAX: 022501; RRID: IMSR_JAX:022501; The Jackson Laboratory) were crossed to *Rosa26*[tdTomato] mice (JAX: 007914; RRID: IMSR_JAX:007914; The Jackson Laboratory) to generate embryos for neural crest-derived sphere culture and for immunofluorescence experiments. Timed pregnancies were used to collect embryonic samples. For embryonic staging, female mice were plug-checked in the morning and those with a vaginal plug were assigned E0.5. Female mice continued to receive standard chow (LabDiet PicoLab Rodent Diet 20) and water *ad libitum* during pregnancy, unless otherwise specified. For embryo collection, pregnant female mice were anesthetized with isoflurane and immediately euthanized by cervical dislocation followed by decapitation. For postnatal staging (i.e. skeletal preparations, μCT scanning), the day of birth was assigned P0. All mouse embryos and pups were genotyped for sex and results are reported for both sexes.

### Mouse handling

Depletion of CSF1R[+] cells was achieved by administering the Plexxikon CSF1R inhibitor PLX5622 (1200 PPM added to chow AIN-76A, Research Diets) to pregnant female mice starting at E3.5. Control dams received control diet (AIN-76A, Research Diets). Pregnant female mice were exposed to either control diet or PLX5622 diet from E3.5 until sample collection. Following birth, female mice were placed on our standard chow (LabDiet PicoLab Rodent Diet 20) from P0 to P1.

### Genotyping

Tails were collected from all mouse embryos/pups and incubated in extraction buffer (100 mM NaCl, 50 mM Tris, 100 mM EDTA, 1% SDS) with proteinase K (8 units, ~0.4 mg/ml; New England Biolabs, P8107S;

UniProtKB: P06873) overnight at 56°C. DNA was extracted from digested tails by precipitation with saturated NaCl solution (~6 M), followed by centrifugation at 20,000 *g* for 10 min. DNA was precipitated from the supernatant with isopropanol, and pellets were collected by centrifugation at 20,000 *g* for 5 min. Pellets were washed with 70% ethanol and dried at 37°C. Dried DNA was dissolved in 200 μl of Tris-EDTA buffer (10 mM Tris, 1 mM EDTA, pH 8) for 1 h at 68°C, and 1.6 μl of DNA was subsequently added to OneTaq® Quick-Load® 2X Master Mix with Standard Buffer (New England Biolabs, M0486S) along with 1 μM each of primers for SX (*Sly* and *Xlr*) (McFarlane et al., 2013), *GFP* (Tiscornia et al., 2003), *Wnt1^{Cre}* (The Jackson Laboratory) or *Rosa26^{tdTomato}* (The Jackson Laboratory) (see Table S5 for primer sequences). PCR reactions were run on a Applied Biosystems™ MiniAmp™ Thermal Cycler (Applied Biosystems) as follows: 95°C for 3 min, 35 cycles of denaturation at 95°C for 30 s, annealing at 60°C (57°C for SX primers) 30 s, and extension at 72°C for 30 s, followed by final extension at 72°C for 5 min. Amplified fragments were run on a 1.5% agarose gel in TAE buffer (0.4 M Tris, 0.2 M acetic acid, 10 mM EDTA, pH 8) and visualized using SmartGlow Pre-stain for nucleic acid gels (Accuris Instruments, E4500-PS) on a Fisherbrand™ Real Time Electrophoresis System (Fisher Scientific). DNA fragments were sized according to GeneRuler Ready-to-Use 100 bp DNA Ladder (Thermo Fisher Scientific, SM0243).

### Flow cytometry
Flow cytometry methods were adapted from Rosin et al. (2018). E11.5, E13.5, E15.5 and E17.5 mouse embryo heads were collected from control and PLX5622-exposed *Csf1r^{EGFP}* mice in PBS on ice. Craniofacial tissues were then micro-dissected away from the whole head while in PBS on ice. The micro-dissected craniofacial tissues were cut/dissociated into smaller pieces using a blade while in culture media on ice containing (v/v): 65.3% DMEM (Gibco 11965-092, Thermo Fisher Scientific), 32.7% F-12 (Gibco 11765-054, Thermo Fisher Scientific), and 2% B-27 supplement (Gibco 17504-044, Thermo Fisher Scientific). The resulting dissociated tissue was filtered through a 35 μm strainer (Falcon, 352235), placed in a chilled 1.5 ml tube and centrifuged at 300 *g* for 5 min at room temperature (RT). The resulting cell pellets were re-suspended in 500 μl cold HBSS with 5% fetal bovine serum (FBS) and filtered through a 35 μm strainer before flow cytometry. The resulting cell suspensions were analyzed by the University of British Columbia's Flow Core Facility using a Beckman Coulter Life Sciences CytoFLEX LX machine and Beckman Coulter CytExpert software (Beckman Coulter Life Sciences, 2025; RRID:SCR_017217).

### Tissue section immunohistochemistry
E15.5 *Csf1r^{EGFP}*, *Wnt1^{Cre}*;*Rosa26^{tdTomato}* and CD1 mouse embryo heads were collected in ice-cold PBS and fixed in 4% paraformaldehyde (PFA) overnight at 4°C. The tissues were then washed in PBS and equilibrated in 20% sucrose/PBS overnight at 4°C. Heads were embedded in Clear Frozen Section Compound (VWR, 95057-838) and cryosectioned (14 μm sections) on a Leica CM1950 cryostat. Cryosections were rehydrated in PBS, washed four times for 10 min each wash with PBT (PBS with 0.1% Triton X-100), permeabilized for 30 min with 1% Triton X-100 in PBS, blocked using 5% normal donkey serum (Sigma-Aldrich) in PBT for 1 h at RT, and exposed to sheep anti-CSF1R (1:200; R&D Systems, AF3818; RRID: AB_884158), rabbit anti-cathepsin K (1:200; Abcam, ab19027; RRID: AB_2261274), rabbit anti-active caspase 3 (1:500; BD Pharmingen, 559565; RRID: AB_397274), rabbit anti-Sp7 (1:500; Abcam, ab209484; RRID: AB_2892207), mouse 2H3 (1:200; Developmental Studies Hybridoma Bank, 2H3; RRID: AB_531793), mouse MF20 (1:100; Developmental Studies Hybridoma Bank, MF 20; RRID: AB_2147781), goat anti-CCL3 (1:200; R&D Systems, AF-450-NA; RRID: AB_354492), goat anti-CCL4 (1:200; R&D Systems, AF-451-NA; RRID: AB_2071055) and/or mouse anti-Ki67 (1:200; BD Pharmingen, 556003; RRID: AB_396287) at 4°C overnight. Slides were then washed four times for 10 min each wash with PBT and exposed to secondary antibody [1:200; Alexa 488 donkey anti-rabbit IgG (Invitrogen, A21206), Alexa 488 donkey anti-mouse IgG (Invitrogen, A21202), Alexa 594 donkey anti-sheep IgG (Invitrogen, A11016), Alexa 594 donkey anti-rabbit IgG (Invitrogen, A21207), Alexa 594 donkey anti-mouse IgG (Invitrogen, A21203) or Alexa 594 donkey

anti-goat IgG (Invitrogen, A11058)] for 2 h at RT. Nuclei were stained with 1:1000 Hoechst 33342 (Invitrogen, H3570; CAS: 23491-52-3) in PBS for 5 min at RT and washed three times for 5 min each wash with PBT. Sections were mounted using Aqua Poly/Mount (Polysciences Inc.). Fluorescent images were captured on a ZEISS Axioplan fluorescent microscope with ZEISS Axiocam HRm camera or a ZEISS LSM900 confocal microscope and further processed using LSM+. Brightness and/or contrast of the entire image was adjusted using ZEISS ZEN 3.9 (RRID:SCR_013672; ZEISS Microscopy, 2025) and/or Adobe Photoshop CC (RRID:SCR_014199; Adobe Inc, 2025) if deemed appropriate.

### Tartrate-resistant acid phosphatase staining
E15.5 *Csf1r^{EGFP}* mouse embryo head cryosections (described above) were rehydrated in distilled water and incubated for 30 min at 37°C in pre-warmed TRAP staining solution consisting of 0.1 mg/ml naphthol AS-MX phosphate (Sigma-Aldrich, N4875; CAS: 1596-56-1), 0.6 mg/ml Fast Red Violet LB Salt (Sigma-Aldrich, F3381; CAS: 32348-81-5), 11.4 mg/ml L-(+) tartaric acid (Sigma-Aldrich, T109; CAS: 87-69-4) and 9.2 mg/ml sodium acetate anhydrous (Sigma-Aldrich) in 0.28% glacial acetic acid, with pH adjusted to 4.7-5.0 with 5 M sodium hydroxide (Sigma-Aldrich). Slides were then washed with distilled water, and sections were mounted using Aqua Poly/Mount (Polysciences Inc.). Light microscope images were captured on a ZEISS Axiocam 208 color camera mounted on a ZEISS Stemi 508 microscope. Brightness and/or contrast of the entire image was adjusted using Adobe Photoshop CC if deemed appropriate.

### Whole-mount immunohistochemistry
Whole-mount staining methods were adapted from Rosin et al. (2015). Antibodies recognizing neurofilament (2H3) and muscle myosin (MF20) were used to visualize embryonic nerves and muscles. E11.5, E12.5 and E13.5 whole CD1 and C57BL/6 mouse embryos were fixed overnight in Dent's fixative (4:1 methanol:DMSO) at 4°C. Embryos were bleached overnight in 5:1 Dent's fixative:30% $H_2O_2$ at RT, then rehydrated with successive 30-min washes in 50%/15%/0% methanol in 0.5% Tween 20 in PBS (PBST). Embryos were blocked twice for 1 h each incubation in PBST/ 1% DMSO/2% skim milk powder (PBSTMD) at RT then incubated overnight at 4°C with primary antibody diluted 1:130-1:150 in PBSTMD to achieve a final concentration of 2 μg/ml. Embryos were then washed four times for 1 h each wash with PBST, blocked with PBSTMD, and incubated overnight at 4°C with peroxidase-conjugated goat anti-mouse IgG (Sigma-Aldrich, A-9169) diluted 1:300 in PBSTMD, then washed four times for 1 h each wash with PBST, stained with 0.5 mg/ml DAB (3,3′-diaminobenzidine tetrahydrochloride; Sigma-Aldrich, D5905; CAS: 7411-49-6) and 0.3% $H_2O_2$ in PBST, dehydrated to 100% methanol, and cleared in BABB (1:2, benzyl alcohol:benzyl benzoate). Light microscope images were captured on a ZEISS Axiocam 208 color camera mounted on a ZEISS Stemi 508 microscope. Brightness and/or contrast of the entire image was adjusted using Adobe Photoshop CC if deemed appropriate.

### Von Kossa staining
Von Kossa staining was adapted from Tosun et al. (2022). E15.5 CD1 mouse embryo head cryosections (described above) were rehydrated in distilled water and incubated in 0.45 μm-filtered 2% silver-nitrate solution (Sigma-Aldrich, 209139; CAS: 7761-88-8) under direct light exposure (14W, 1500 lumen bulb) for 1 h, washed with 1% acetic acid, and subsequently stained for 20 min in 0.02% Alcian Blue in 70% ethanol and 30% acetic acid. Sections were counterstained in 0.1% Nuclear Fast Red Solution for 10 min and rinsed in distilled water before mounting with Aqua Poly/Mount (Polysciences Inc.). Slides were imaged on a ZEISS Axiocam 208 color camera mounted on a ZEISS Stemi 508 microscope.

### Micromass culture
Micromass culture methods were adapted from Ralphs (1992), Ueharu et al. (2022) and Tophkhane et al. (2024). E12.5 CD1 mouse embryo heads were collected and the frontonasal mass and mandible were dissected in cold Hank's balanced saline solution (HBSS, without calcium and magnesium; Thermo Fisher Scientific, 14185052) with 10% FBS and 1% penicillin/ streptomycin. Dissected craniofacial tissues were individually incubated in

2% trypsin (Gibco) on ice for 1 h. HBSS with 10% FBS and 1% penicillin/streptomycin was added to inhibit the enzymatic activity of trypsin, pipetting up and down to dissociate the ectoderm and mesenchyme. The cell solution was then centrifuged at 1000 $g$ for 5 min. The supernatant was removed, and the cells were resuspended in HBSS with 10% FBS and 1% penicillin/streptomycin. The resulting suspension was filtered through a 35 µm strainer. The mesenchymal cells were counted using a Countess II automated cell counter (Thermo Fisher Scientific) and were resuspended to $2 \times 10^7$ cells/ml in chondrogenic medium containing 40% DMEM (Gibco 11995-065, Thermo Fisher Scientific) and 60% F12 (Gibco 11765-054, Thermo Fisher Scientific) supplemented with 10% FBS, GlutaMAX (Gibco, 35050061), 50 µg/ml ascorbic acid (Thermo Fisher Scientific, 850-3080IM; CAS: 50-81-7), 10 mM β-glycerol phosphate (Sigma-Aldrich, G9422; CAS: 154804-51-0) and 1% penicillin/streptomycin. Cells were dropped into Nunc cell culture-treated 24-well plates (Thermo Fisher Scientific, 142475) in 10 µl drops at $2 \times 10^5$ cells/well in the center of the well and incubated at 37°C and 5% $CO_2$ for 90 min to allow cells to attach and then flooded with 1 ml/well of chondrogenic medium. The culture medium was changed every other day (days 3, 5 and 7) until day 8. On day 8, cultures were fixed in 4% PFA for 30 min at RT and stored at 4°C in 100 mM Tris (pH 8.3) until staining. Fixed cultures were incubated at RT in 0.6 mg/ml Fast Red Violet LB salt and 0.1 mg/ml naphthol AS-MX phosphate in 100 mM Tris (pH 8.3) for 60 min to stain for alkaline phosphatase activity. The cultures were then stained with 1% Alcian Blue in 3% acetic acid and 1% HCl to detect the area occupied by cartilage. All cultures were counterstained with 50% Shandon's Instant Hematoxylin (Thermo Fisher Scientific, 6765015) and stored in 100% glycerol. Stained cultures were visualized on a ZEISS Stemi 508 light microscope, and images were captured on a ZEISS Axiocam 208 color camera.

### Bone and cartilage skeletal staining
P1 CD1 and C57BL/6 mouse pups were collected and euthanized by decapitation. Eyes were removed from skinned heads, and heads were then fixed in ethanol with 1% glacial acetic acid for at least 24 h and placed in 0.45 µm-filtered Alcian Blue solution (1 mg/ml Alcian Blue 8GX in 80% ethanol and 20% glacial acetic acid; Sigma-Aldrich, 05500; CAS: 33864-99-2) overnight to stain cartilage. After washing with ethanol for two 1 h washes, mouse heads were transferred to 1.5% KOH solution for 3.5 h, then stained with Alizarin Red S solution (0.15 mg/ml Alizarin Red S in 0.5% KOH; Sigma-Aldrich, A5533; CAS: 130-22-3) for 3 h to stain bone. Skulls were cleared by incubation in 0.5% KOH with 20% glycerol twice for 2-3 days each incubation and stored in a 40% glycerol mixture. Light microscope images were captured on a ZEISS Axiocam 208 color camera mounted on a ZEISS Stemi 508 microscope. Brightness and/or contrast of the entire image was adjusted using Adobe Photoshop CC if deemed appropriate.

### µCT
P1 CD1 mouse pups were collected and euthanized by decapitation. Heads were frozen at −20°C in 15 ml Falcon tubes. For contrast enhanced µCT scanning, P1 heads were fixed by overnight incubation with 4% PFA at 4°C. The samples were then dehydrated through serial incubations with increasing methanol concentrations (30%, 50% and 70%) in PBS. Subsequently, samples were incubated in a 1% phosphotungstic acid/90% methanol solution for 2 weeks, ending with rehydration through serial incubations with decreasing methanol concentrations (70%, 50% and 30%). Incubation with each methanol concentration at both the dehydration and rehydration steps was done for 2 days. µCT scans were performed with the Scanco Medical µCT100 scanner at 55 kVp and 200 µA in the Centre for High-Throughput Phenogenomics at the University of British Columbia. Phosphotungstic acid-stained specimens were scanned at 7 µm resolution while embedded in 1% agarose. Additionally, unstained heads were µCT-scanned after thawing at RT for 1 h, at 15 µm resolution while embedded in 1% agarose. All outputs were exported as DICOM files.

### Craniofacial tissue culture for cytokine and chemokine analysis
Methods for tissue culture for cytokine/chemokine analysis were adapted from Rosin et al. (2021). E13.5 C57BL/6 mouse embryo heads were collected in PBS at RT. Craniofacial tissues were then micro-dissected away

from the whole head and placed in a cell culture dish with 400 µl of 37°C culture media containing (v/v): 65.3% DMEM, 32.7% F-12 and 2% B27 supplement. Craniofacial tissues were cultured in a humid incubator with ambient oxygen and 5% $CO_2$ at 37°C for 6 h. Culture media was collected and spun down at 3000 $g$ for 10 min at RT, then supernatant was collected into a new tube, flash-frozen, and stored at −75°C until sent for multiplexed quantification of 45 mouse cytokines, chemokines and growth factors using Luminex xMAP technology with MILLIPLEX® Mouse Cytokine/Chemokine Magnetic Bead Panel (Millipore, MCYTOMAG-70K) and MILLIPLEX® Mouse Cytokine/Chemokine Magnetic Bead Panel II (Millipore, MECY2MAG-73K), according to the manufacturer's protocol. The analytes were evaluated on the Luminex™ 200 system by Eve Technologies Corp. (Calgary, AB, Canada).

### Single-cell RNA sequencing data analysis
Publicly available single-cell RNA sequencing datasets of wild-type E12.5 and E13.5 C57BL/6J and FVB/NJ mouse craniofacial mesenchyme were downloaded from the NCBI Gene Expression Omnibus database (Angelozzi et al., 2022; Rajderkar et al., 2024). Accession numbers GSM5324643 (E13.5 C57BL/6J), GSM6146223 (E12.5 C57BL/6J) and GSM7508877 (E12.5 FVB/NJ) were imported into Seurat v5 (RRID: SCR_016341; Hao et al., 2024, 2021; Stuart et al., 2019; Butler et al., 2018; Satija et al., 2015) in RStudio (RRID:SCR_000432; Posit team, 2025), which runs on the R Project for Statistical Computing software environment (RRID:SCR_001905; R Core Team, 2024), for analysis. Low-quality cells (>20,000 counts, >6000 genes detected or >20% mitochondrial genes for GSM6146223; >35,000 counts, >7000 genes detected or >15% mitochondrial genes for GSM5324643; <1500 or >7500 genes detected for GSM7508877) were filtered from downstream analysis as they may have arisen from damaged cells, multiplets, or other processing artifacts. Counts were normalized and datasets were filtered to leave only cells with non-zero *Csf1r* expression. *Csf1r*-expressing cell clusters were identified using graph-based clustering with the 'FindClusters' function, based on variable features and similar cells identified by the 'FindNeighbors' function. The 'clustree' package was used to assist in determining 0.2 as the appropriate resolution for clustering, and proper clustering was then verified by manual inspection of marker genes. Uniform manifold approximation and projections (UMAPs) were visualized using 'DimPlot' and gene signatures were visualized using 'Vlnplot'. A cluster of hemoglobin-enriched cells did not express any markers of monocytes, microglia, macrophages or osteoclasts except *Csf1r*; this cluster was excluded in the final analysis. All source code for single-cell RNA sequencing data analysis and descriptions of functions can be found on GitHub at https://github.com/RosinLabUBC/scRNAseq-code-FM.

### Fluorescence *in situ* hybridization
E15.5 CD1 mouse embryo head cryosections (described above) were rehydrated in PBS and stained according to the manufacturer's protocol using the RNAscope™ Multiplex Fluorescent Reagent Kit v2 (Advanced Cell Diagnostics, 323100). Briefly, sections were post-fixed for 15 min at 4°C in 4% PFA. Slides were then incubated in hydrogen peroxide for 10 min, washed with water, and incubated in boiling target retrieval buffer for 5 min. Sections were washed with water and dehydrated in 100% ethanol for 3 min and treated with protease 3 for 30 min at 40°C in a ACD HybEZ™ II Hybridization System (Advanced Cell Diagnostics). Slides were washed with water and probes for *Csf1r* (Advanced Cell Diagnostics, 428191-C3) and *Ccl4* (Advanced Cell Diagnostics, 421071) were hybridized to tissue for 2 h at 40°C, followed by incubation in amplification reagents and TSA Vivid 520 (Advanced Cell Diagnostics, 323271) and TSA Vivid 570 (Advanced Cell Diagnostics, 323272) fluorophores, with washes in wash buffer between incubations. Nuclei were stained with DAPI for 30 s and slides were mounted using Aqua Poly/Mount (Polysciences Inc.). High-resolution images were captured on a ZEISS LSM900 confocal microscope and were further processed using LSM+. Brightness and/or contrast of the entire image was adjusted using ZEISS ZEN 3.9 and/or Adobe Photoshop CC if deemed appropriate.

### NCC sphere culture
NCC sphere culture methods were adapted from Rosin et al. (2021) and Hagiwara et al. (2014). E12.5 mouse heads were collected (*Wnt1^Cre* mice

crossed to *Rosa26^tdTomato* mice to label neural crest-derived cells with tdTomato) in 37°C PBS containing 1% penicillin/streptomycin. Craniofacial tissues were then micro-dissected away from the whole head while in PBS containing 1% penicillin/streptomycin. Micro-dissected craniofacial tissues were cut/dissociated into smaller pieces using a sterile blade. The resulting dissociated tissue was filtered through a 35 μm strainer, placed in a 1.5 ml tube and centrifuged at 300 *g* for 10 min at RT. The resulting cell pellets were re-suspended in 500 μl of 37°C warmed culture media containing (v/v): 98% DMEM/F-12 (no Phenol Red; Gibco 21041-025, Thermo Fisher Scientific) and 2% B-27 supplement, with 20 ng/ml each of human recombinant epidermal growth factor (STEMCELL Technologies, 78006.1) and basic fibroblast growth factor (STEMCELL Technologies, 78003.1) added. The resulting cell suspensions were filtered through a 35 μm strainer before fluorescence-activated cell sorting by the University of British Columbia's Flow Core Facility using a Beckman Coulter MoFlo Astrios EQ cell sorter. tdTomato⁺ neural crest-derived cells were collected in 500 μl of culture media. Cells were plated at 12,500 cells/ml per well in 24-well plates and were incubated for 10 days at 37°C with 5% CO₂, with 50% media replenishment at day 5. At day 10, primary spheres were counted and imaged, dissociated in sterile media, counted and re-plated at 1000 cells/ml and cultured again for 10 days as above for secondary sphere formation.

### Neural crest sphere immunohistochemistry

Cultured neural crest-derived spheres (described above) were fixed in 4% PFA for 30 min at RT and stored in PBS at 4°C until staining. Spheres were washed with PBT (PBS with 0.1% Triton X-100), blocked using 5% normal donkey serum (Sigma-Aldrich) for 1 h at RT, and exposed to goat anti-PDGFRα (1:150; R&D Systems, AF1062; RRID: AB_2236897), rabbit anti-SOX2 (1:500; Millipore, AB5603; RRID: AB_2286686) or mouse anti-SOX10 (1:500; R&D Systems, MAB2864; RRID: AB_2195180) at 4°C overnight. Spheres were then washed with PBT and exposed to secondary antibody [1:200; Alexa 488 donkey anti-rabbit IgG (Invitrogen, A21206), Alexa 488 donkey anti-mouse IgG (Invitrogen, A21202) or Alexa 488 donkey anti-goat IgG (Invitrogen, A11055)] for 2 h at RT. Nuclei were stained with Hoechst 33342 (Invitrogen) for 5 min at RT. Spheres were aspirated with a wide bore pipette tip and deposited onto a glass microscope slide, and were subsequently mounted using Aqua Poly/Mount (Polysciences Inc.). High-resolution images were captured on a ZEISS LSM900 with confocal microscope and were further processed using LSM+. Brightness and/or contrast of the entire image was adjusted using ZEISS ZEN 3.9 and/or Adobe Photoshop CC if deemed appropriate.

### Phagocytosis assay

E13.5 mouse embryo heads were collected from *Csf1r^EGFP* mice in 37°C PBS. *Csf1r^EGFP* embryos were screened for EGFP expression on a ZEISS Axiovert 5 microscope. *Csf1r^EGFP+* craniofacial tissues were then micro-dissected away from the whole head while in 37°C PBS. The micro-dissected craniofacial tissues were bisected along the midline using a blade, and each half was cut once more into rostral and caudal quarters. Each half-head (one rostral quarter plus one caudal quarter of the same side) was placed in 300 μl of 37°C culture media containing (v/v): 60.7% DMEM (Gibco 11965-092, Thermo Fisher Scientific), 30.3% F-12 (Gibco 11765-054, Thermo Fisher Scientific), 2% B-27 supplement (Gibco 17504-044, Thermo Fisher Scientific), 7% PBS, 10 ng/ml CCL2 (R&D Systems, 479-JE), 20 ng/ml CCL3 (R&D Systems, 450-MA), 40 ng/ml CCL4 (R&D Systems, 451-MB), 20 ng/ml CCL12 (R&D Systems, 428-P5), 30 ng/ml CXCL2 (R&D Systems, 452-M2), 30 ng/ml CXCL10 (R&D Systems, 466-CR) and 20 ng/ml IL6 (R&D Systems, 406-ML). Each contralateral half-head was placed in 300 μl of identical culture media without cytokines. All craniofacial tissues were incubated at 37°C with 5% CO₂ for 3 h. pHrodo™ Deep Red *Escherichia coli* BioParticles™ Conjugate (Invitrogen P35360, Thermo Fisher Scientific) were warmed and homogenized by addition of 171.4 μl of 37°C culture media to 400 μl of BioParticles, then 50 μl of 37°C BioParticles-media was added to each 300 μl craniofacial culture. Craniofacial cultures were further incubated at 37°C with 5% CO₂ for 1 h. After 1 h incubation, craniofacial tissues were removed from culture media with fine forceps and placed in 3 cm Petri dishes containing 37°C HBSS with 5% FBS. The craniofacial tissues were cut/dissociated into smaller

pieces using a blade, filtered through a 35 μm strainer (Falcon 352235), placed in a 1.5 ml tube and centrifuged at 300 *g* for 5 min at RT. The resulting cell pellets were re-suspended in 500 μl 37°C HBSS with 5% FBS and filtered through a 35 μm strainer before flow cytometry. The resulting cell suspensions were analyzed by the University of British Columbia's Flow Core Facility using a Beckman Coulter Life Sciences CytoFLEX LX machine and Beckman Coulter CytExpert software.

### Quantification and statistical analysis

E15.5 craniofacial tissue sections were obtained by cryosectioning PFA-fixed frozen embryo heads on a Leica CM1950 cryostat. Tissue was sectioned coronally starting from the nose, and 14 μm serial tissue sections were captured across ten slide sets, with 21 embryonic head sections placed on each slide, and each set comprising two slides. A single set (two slides) of serial sectioned slides was used for each staining experiment described above. Every 14 μm section containing the craniofacial structure of interest was used for cell counts of cells with positive expression of *Csf1r^EGFP*, CSF1R, cathepsin K, cleaved caspase 3, *Wnt1^Cre;Rosa26^tdTomato* and Ki67 (*n*=3 per sex/treatment from two or three dams). For CSF1R, *Csf1r^EGFP*, cathepsin K and cleaved caspase 3 counts, positive cells were counted within the ear, eye, trigeminal, tongue and mandible. A 5-cell radius was used for counting cells surrounding the nasal septum, Meckel's cartilage and maxillary incisor. For *Wnt1^Cre;Rosa26^tdTomato* counts, the mineralizing region of the premaxilla and mandible were outlined and cells within this area were counted. For Ki67 counts, mineralizing areas of the premaxilla and mandible were outlined comparably to the *Wnt1^Cre; Rosa26^tdTomato* outlines, and Ki67-positive cells within this area were counted. For micromass alkaline phosphatase staining analysis, images were adjusted by Fiji v2.16 (RRID:SCR_002285; Schindelin et al., 2012) with ImageJ v1.54 (RRID:SCR_003070; Schneider et al., 2012) with the 'Color Threshold' function using red, green and blue (RGB) color space (Schindelin et al., 2012; Schneider et al., 2012; Gutiérrez et al., 2012). The mask created by the threshold color was set to black and white (B&W). The image was converted to binary by setting image type to '8-bit'. The stained area was first delineated using the 'Freehand selections' tool, and the 'Edit Clear Outside' function was used to eliminate pixels outside of the selected area. 'Analyze Particles' was used to measure the total stained area. The process was repeated for Hematoxylin-stained images to obtain the total micromass culture area. The same process was also used to analyze the TRAP⁺ area in the premaxilla, maxilla and mandible, and for analyzing the von Kossa-stained mineralized area in the maxilla for craniofacial tissue sections. For analysis of the premaxilla and mandible of von Kossa-stained sections, images were imported into Fiji v2.16 (RRID:SCR_002285; Schindelin et al., 2012) with ImageJ v1.54 (RRID:SCR_003070; Schneider et al., 2012) and the unstained area within each mineralized structure was measured with the 'Freehand selections' and 'Measure' tools. For neural crest-derived sphere analysis, each well was quantified by visualization on a ZEISS Axiovert 25 at 10× magnification, and a minimum cell cluster diameter of 40 μm was used as a cut-off to qualify as a counted sphere. Quantitative results (*n*=3-6 mouse embryos from two to four dams, unless otherwise mentioned in the Results and/or figure legends) for all counts and measurements are represented by mean±s.e.m. and were either analyzed by a two-tailed, unpaired *t*-test, two-way ANOVA with Tukey's post hoc analysis in GraphPad Prism 9 (RRID:SCR_002798; GraphPad Software, 2025) or an aligned ranks transformation ANOVA with Tukey's post hoc analysis using ARTool v0.11.2 (RRID:SCR_027412; Kay et al., 2025, Wobbrock et al., 2011) in RStudio (RRID:SCR_000432; Posit team, 2025), which runs on the R Project for Statistical Computing software environment (RRID:SCR_001905; R Core Team, 2024). Statistical significance was defined as $P<0.05$.

### Acknowledgements

We thank Andrew Johnson and Justin Wong from the UBC Flow Core Facility for help with fluorescence-activated cell sorting, the UBC Centre for High-Throughput Phenogenomics (a facility supported by the Canada Foundation for Innovation, British Columbia Knowledge Development Foundation and the UBC Faculty of Dentistry) for help with μCT scans, and the UBC Centre for Disease Modeling for animal care. We would also like to acknowledge Shruti S. Tophkhane, Joaquin I. Henriquez, Katherine Fu, Joy M. Richman and the entire Richman Lab for help with micromass culture and TRAP staining, and Siddharth R. Vora for help with the μCT scans.

## Competing interests

The authors declare no competing or financial interests.

## Author contributions

Conceptualization: F.M., J.M.R.; Data curation: F.M., M.R., R.O.M.; Formal analysis: F.M., R.O.M., J.M.R.; Funding acquisition: J.M.R.; Investigation: F.M., R.R.J.Z., M.R., I.Z., S.O., R.O.M., V.B.W., J.M.R.; Methodology: F.M., M.R., R.O.M., J.M.R.; Project administration: J.M.R.; Resources: F.M., M.R., J.M.R.; Software: F.M., R.O.M.; Supervision: J.M.R.; Validation: F.M., M.R., R.O.M., J.M.R.; Visualization: F.M., M.R., R.O.M., J.M.R.; Writing – original draft: F.M., J.M.R.; Writing – review & editing: F.M., M.R., J.M.R.

## Funding

F.M. was supported by a Natural Sciences and Engineering Research Council of Canada (NSERC) PGS-D (579343-2023), a University of British Columbia (UBC) Four Year Doctoral Fellowship, and a UBC Faculty of Dentistry Joseph Tonzetich Fellowship. R.R.J.Z. was supported by a UBC Faculty of Dentistry Summer Research Award (2022 and 2023) and a Charles Shuler Research Fellow award (2023). I.Z. and S.O. were supported by a UBC Faculty of Dentistry Summer Research Award (2025). V.B.W. was supported by an NSERC USRA. This work was supported by an NSERC Discovery Grant to J.M.R. (RGPIN-2022-03718). J.M.R. is a Michael Smith Health Research BC Scholar and a Tier 2 Canada Research Chair (Canada Research Chairs) in Immune Regulation of Developmental Programs. Open Access funding provided by Natural Sciences and Engineering Research Council of Canada. Deposited in PMC for immediate release.

## Data and resource availability

All source code for single-cell RNA sequencing data analysis and descriptions of functions can be found on GitHub at https://github.com/RosinLabUBC/scRNAseq-code-FM. Images of control and PLX5622-exposed whole-mount E11.5, E12.5 and E13.5 nerve (2H3) and muscle (MF20) staining and P1 skeletal staining described in this article are available at FaceBase (https://doi.org/10.25550/A9-12PJ; Ma and Rosin, 2026). All other relevant data and details of resources can be found within the article and its supplementary information.

## Peer review history

The peer review history is available online at https://journals.biologists.com/dev/lookup/doi/10.1242/dev.205423.reviewer-comments.pdf

## Special Issue

This article is part of the Special Issue 'The Extracellular Environment in Development, Regeneration and Stem Cells', edited by Alex Hughes and Rashmi Priya. See related articles at https://journals.biologists.com/dev/issue/153/16.

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
