## [Peer Review File · Development (Cambridge, England)]

CSF1R+ macrophage and osteoclast depletion impairs neural crest proliferation and craniofacial morphogenesis

Felix Ma, Rose Ru Jing Zhou, Matthew Rosin, Iris Zhou, Sabrina Ownsworth, Rouzbeh Ostadsharif Memar, Vincent B. Wong and Jessica M. Rosin
DOI: 10.1242/dev.205423

Editor: James Briscoe

Review timeline

Original submission:	4 December 2025
Editorial decision:	29 December 2025
First revision received:	16 February 2026
Accepted:	3 March 2026

Original submission

First decision letter

MS ID#: dev.205423

MS TITLE: CSF1R+ macrophage and osteoclast depletion impairs neural crest proliferation and craniofacial morphogenesis

AUTHORS: Felix Ma, Rose Ru Jing Zhou, Matthew Rosin, Iris Zhou, Sabrina Ownsworth, Rouzbeh Ostadsharif Memar, Vincent B. Wong and Jessica M. Rosin

Dear Dr Rosin,

I have now received all the referees' reports on the above manuscript, and have reached a decision. The referees' comments are appended below.

As you will see, the referees express considerable interest in your work, but have some significant criticisms and recommend a substantial revision of your manuscript before we can consider publication. The referees recognise the significance of your contribution to understanding CSF1R signalling in craniofacial development but require revisions to strengthen the manuscript. Most importantly you should address the incomplete CSF1R+ cell depletion (30-50% reduction) and provide direct comparisons with published genetic knockout data. The referees request that you add quantitative analyses to Figures 2, 3, and 4, and implement consistent stratification of male versus female data throughout the manuscript with explicit justification when pooling data. The referees also suggest additional clarifications that would strengthen the study.

If you are able to revise the manuscript along the lines suggested, which may involve further experiments, I will be happy to receive a revised version of the manuscript. Your revised paper will be re-reviewed by one or more of the original referees, and acceptance of your manuscript will depend on your addressing satisfactorily the reviewers' major concerns. Please also note that Development will normally permit only one round of major revision. If it would be helpful, you are welcome to contact us to discuss your revision in greater detail. Please send us a point-by-point response indicating your plans for addressing the referees' comments, and we will look over this and provide further guidance.

Please attend to all of the reviewers' comments and ensure that you upload both a 'clean' version of your Word file, along with a highlighted version clearly showing where you have made changes in the revised manuscript. Please avoid using 'Tracked changes' in Word files as these are lost in PDF conversion. I should be grateful if you would also provide a point-by-point response detailing how you have dealt with the points raised by the reviewers in the 'Response to Reviewers' box. If you do not agree with any of their criticisms or suggestions please explain clearly why this is so.

Reviewer 1

Advance summary and potential significance to field

I found this ms very interesting. The Introduction sets out clearly the background of the study and how it builds on their previous observation that *Csf1r* knockout mice (hence deficient CSF1R signalling) show major effects on craniofacial development. Since CSF1R signalling is essential for macrophage and osteoclast proliferation, differentiation and survival, further work was justified to reveal details of the roles of these cell types in craniofacial development. Osteoclast activity is an important factor in remodelling of the skull during both prenatal development and postnatal growth, so its investigation in this study it is very welcome. A pharmacological inhibitor, PLX5622 was used to suppress the activities of the receptor more flexibly and specifically than was possible by studying the knockout. The authors used an impressively wide range of investigative techniques to analyse the effects of disrupting CSF1R signalling; their thoroughness is also evident in that they compared the effects of this disruption on males and females, and between two different strains of mice. The results are clearly presented and well-illustrated. This study makes a significant contribution to the field of craniofacial development.

I would like to thank the authors for a valuable contribution to the field of craniofacial development and for all the time, care and expertise they have invested in it.

Comments for the author

I have a few suggestions to improve the clarity of the presentation. First, it is many years since I used GFP, so I was baffled by the sudden introduction at the beginning of the Results section (L117) of *Csf1rEGFP* transgenic embryos. There are clues in a few places, e.g. L132, but the first straightforward statement that this is a transgenic fluorescent reporter mouse, enabling CSF1R+ cells to be assessed by immunohistochemistry, appears at the start of the Discussion (L477). This is too late - it would be kind to ignorant readers like me to move this statement to the first mention at the start of the Results.

I found the discussion section on neural crest less intellectually satisfying than the other sections. L592 states specifically that craniofacial nerves arise from NCCs. The two references cited here are 164 and 165. 164 is an avian study, which is unsuitable and unnecessary given the wealth of mammalian data. Apart from the fact that NCC contribute only to the sensory component of the nerves, the most important error is the failure to acknowledge that cells delaminating from epibranchial placodes and the otic vesicle also make major contributions to the formation of craniofacial nerves, and that NCC differentiate to form glial cells as well as neurons. The otic vesicle is the almost exclusive source of sensory neurons for the vestibulocochlear nerve (VIII) (Washausen and Knabe, *Front Cell Dev Biol* 9, 712522, 2021). PLX5672 clearly has effects on non-neural NC derivatives; its apparent lack of an effect of on cranial nerve development may be due to the variability of the NCC contribution here as well as the likelihood that their commitment to a neuronal phenotype involves a different signalling system from that leading to osteogenesis.

I was unimpressed with the significance given to the *Tcof1* mutant mouse. It felt as if it was just there because the authors felt that something to compare with the PLX5622 NC-derived phenotypic effects should be included. It's a long discussion, and peripheral additions to the main components that seem to be argued into relevance detract from the generally enjoyable flow of logic.

One final comment: given the information that there are strain-related effects, the authors suggest (L 572-575) that further studies on PLX5622 on other strains could be useful. Have they considered how much time (for both experimental work and grant application-writing) this would entail? Would it add significantly to the discoveries of this study, which they must already be planning to follow up?

Reviewer 2*Advance summary and potential significance to field*

In this manuscript, Ma et al. investigate the effects of a PLX5622 diet, a potent CSF1R inhibitor, on craniofacial development, with a particular focus on osteoclast depletion, neural crest cell proliferation, and craniofacial morphological changes across embryonic developmental stages. The study also compares differences between male and female embryos. The results are intriguing, and I have a few suggestions below:

1) Throughout the study, the authors relied solely on a PLX5622-containing diet to treat pregnant mice, resulting in an approximately 30-50% reduction in Csf1rEGFP+ cells in the embryos (Fig. 1B). However, immunofluorescence staining (Fig. 1C-S) shows that CSF1R signals do not fully co-localize with Csf1rEGFP+ signals, raising questions about the reliability of the Csf1rEGFP+ reporter in this context. Moreover, a 30-50% reduction in Csf1rEGFP+ cells does not constitute true depletion of CSF1R+ macrophages. The authors discuss phenotypes in genetic models, including in the limitations section, but perhaps could provide a direct comparison of published data with the phenotypes they obtained as a way to assess their model relative to the gold-standard genetic approach.

2) Some of the data are either not quantified or not quantified appropriately. For instance, most panels in Fig. 2, the entirety of Fig. 3, and large portions of Fig. 4 lack quantitative analyses. In addition, the presentation and quantification of male versus female data are inconsistent: in some bar graphs, male and female data appear to be pooled, whereas in others they are presented separately. This makes it difficult to assess the extent to which sex influences the observed phenotypes. A clear and consistent strategy for data stratification separating male and female data throughout with explicitly justified pooled analyses would enable rigorous evaluation of sex-specific and overall effects.

3) A minor comment is that although the Results section is divided into seven subsections, only Figs. 1 and 2 are used to support the first subsection, while all data corresponding to subsections 2 and 3 are placed entirely in the supplementary figures. In contrast, Figs. 3 and 4 are devoted solely to subsection 4. Perhaps this could be rebalanced to align the figures with the sections.

First revisionAuthor response to reviewers' comments

Reviewer #1 thought our manuscript outlined “*an impressively wide range of investigative techniques to analyze the effects of disrupting CSF1R signalling*”, stating that the results are “*clearly presented and well- illustrated*” and that our study “*makes a significant contribution to the field of craniofacial development*”. **Reviewer #1** appreciated “*the time, care and expertise*” we invested, but did include some suggestions to improve the clarity of the manuscript, which we have responded to below:

Suggestions:

1. “*First, it is many years since I used GFP, so I was baffled by the sudden introduction at the beginning of the Results section (L117) of Csf1rEGFP transgenic embryos. There are clues in a few places, e.g. L132, but the first straightforward statement that this is a transgenic fluorescent reporter mouse, enabling CSF1R+ cells to be assessed by immunohistochemistry, appears at the start of the Discussion (L477). This is too late - it would be kind to ignorant readers like me to move this statement to the first mention at the start of the Results.*”

We apologize for this oversight and have addressed this error in the revised manuscript by now saying: “Craniofacial tissues were collected from *Csf1r^{EGFP}* transgenic embryos, enabling CSF1R-expressing cells to be assessed by EGFP+ signal using flow cytometry (Figure 1A), which revealed...” (lines 118-120), and refer to the use of this transgene and

the quantification of EGFP⁺ cells in all analysis thereafter as “*Csf1r*^{EGFP+} cells”.

2. Point #1: “*I found the discussion section on neural crest less intellectually satisfying than the other sections*”

Point #2: “*I was unimpressed with the significance given to the *Tcof1* mutant mouse. It felt as if it was just there because the authors felt that something to compare with the PLX5622 NC-derived phenotypic effects should be included*”

Point #3: “*It's a long discussion, and peripheral additions to the main components that seem to be argued into relevance detract from the generally enjoyable flow of logic.*”

Point #4: “*One final comment: given the information that there are strain-related effects, the authors suggest (L 572-575) that further studies on PLX5622 on other strains could be useful. Have they considered how much time (for both experimental work and grant application-writing) this would entail? Would it add significantly to the discoveries of this study, which they must already be planning to follow up?*”

We apologize for the lengthy discussion and inclusion of topics surrounding the neural crest that were not intellectually satisfying (points #1, 3), including the commentary on the *Tcof1* mutant mouse (point #2). With respect to point #4, we completely agree with Reviewer 1. It took us a significant amount of time and funding to put together the current manuscript, examining both CD1 and C57 strains, and while it allowed us to compare similarities and/or differences seen in published *Csf1/Csf1r* genetic studies conducted in various strains of rodents (please see Table S3 for a full description), further study of strain differences would not significantly add to the discoveries of this study. Therefore, to address points #1-4, we have removed lines 678-696, 703-715, and 739-752 in the discussion (please see edited version with strikethrough text).

3. “*L592 states specifically that craniofacial nerves arise from NCCs. The two references cited here are 164 and 165. 164 is an avian study, which is unsuitable and unnecessary given the wealth of mammalian data. Apart from the fact that NCC contribute only to the sensory component of the nerves, the most important error is the failure to acknowledge that cells delaminating from epibranchial placodes and the otic vesicle also make major contributions to the formation of craniofacial nerves, and that NCC differentiate to form glial cells as well as neurons. The otic vesicle is the almost exclusive source of sensory neurons for the vestibulocochlear nerve (VIII) (Washausen and Knabe, *Front Cell Dev Biol* 9, 712522, 2021). PLX5622 clearly has effects on non-neural NC derivatives; its apparent lack of an effect of on cranial nerve development may be due to the variability of the NCC contribution here as well as the likelihood that their commitment to a neuronal phenotype involves a different signalling system from that leading to osteogenesis.*”

We would like to thank Reviewer 1 for catching this important point and providing applicable literature for reference. Unfortunately, we have had to remove this section of the text and substantially cut down our Discussion, both to address points made above about the neural crest, and to allow for extensive revision of the results to accommodate the requests of Reviewer 2 (please see below). Given that cranial nerves show no overt differences in PLX5622 embryos up to E15.5, and we recommend future studies to examine craniofacial nerves and muscles postnatally (discussed in lines 550-556), we have removed all further discussion of nerve development.

Reviewer #2 thought our results were “*intriguing*”, and also included some suggestions to improve the clarity of the manuscript, which we have responded to below:

1. “*Throughout the study, the authors relied solely on a PLX5622-containing diet to treat pregnant mice, resulting in an approximately 30-50% reduction in *Csf1r*^{EGFP+} cells in the*

embryos (Fig. 1B). However, immunofluorescence staining (Fig. 1C-S) shows that CSF1R signals do not fully co-localize with Csf1rEGFP+ signals, raising questions about the reliability of the Csf1rEGFP+ reporter in this context.”

We thank Reviewer 2 for this comment. It is considered normal and common for fluorescence transgenes (e.g., EGFP) to not fully overlap with antibody-based protein staining (e.g., immunofluorescence staining). Although the goal of fluorescent transgenes is to mimic endogenous protein, several biological and technical factors can lead to partial mismatches in localization or expression levels, including size of the tag (steric hindrance), overexpression artifacts (i.e., differences in promoters), maturation time, fixation/processing artifacts, and protein turnover mismatch, to name just a few.

Therefore, seeing that “~92% of *Csf1r*^{EGFP+} cells were found to also express CSF1R” (lines 133-134), is quite normal. However, we include both *Csf1r*^{EGFP+} cell (transgene) quantification and CSF1R+ cell (immunofluorescence) quantification in our manuscript (see Figure 1) for this exact purpose, even though they display quite comparable depletion levels across various tissues. Moreover, in the discussion we state: “Although we found EGFP signal in cells negative for CSF1R immunostaining, this could be due to variable timing between *Csf1r*^{EGFP} transgene expression and CSF1R protein or alternatively, EGFP signal could be retained” (lines 545-548), and have now included reference to literature supporting small discrepancies between transgene and antibody-based protein staining.

2. *“A 30-50% reduction in Csf1rEGFP+ cells does not constitute true depletion of CSF1R+ macrophages.*

The authors discuss phenotypes in genetic models, including in the limitations section, but perhaps could provide a direct comparison of published data with the phenotypes they obtained as a way to assess their model relative to the gold-standard genetic approach.”

We thank Reviewer 2 for this comment. Indeed, depletion rates of CSF1R+ cells in PLX5622 embryos vary by method of analysis (whole head assessed via flow cytometry vs. specific structures analyzed using IF staining), sex (males vs. females), and tissue (e.g., nasal septum vs. tongue), which can range between 21-80% depletion (please see Figure 1 and Table S3). However, it should be noted that this is consistent with genetic studies which report depletion of CSF1R+ cells between 26-100% depending on the model (*Csf1* KO vs. *Csf1r* KO, mouse vs. rat, etc.), strain, and tissue analyzed (please see Table S3). It is also important to note that although we may observe milder depletion of macrophages in our model, we also characterize complete absence of embryonic osteoclasts and several robust craniofacial phenotypes that impact 100% of exposed offspring across multiple litters and two strains of mice.

To provide a direct comparison of the results from our study to published data using a genetic approach, we have generated a Table (please see Table S3) that is referenced across the discussion. Comparison of phenotypes across all these models in the text would create a lengthy discussion (and would be a good topic for a review!); therefore, to mitigate the increased text required to address comments below in the results section, we have instead included an extensive comparison of phenotypes in Table format. It is important to note that with respect to published literature using genetic approaches, we have included findings in the Table that we have outlined as “not reported in study by observed in figures” in order to give credit where credit is due, that is, if the study directly showed a phenotype in a Figure but did not report it in the text. From this point of view, you can see that almost all the postnatal phenotypes observed in our study using PLX5622 are consistent with genetic models. If Reviewer 2 does not deem it appropriate to include phenotypes that are not explicitly described in the text, we would be happy to modify Table S3 to only include phenotypes explicitly described by the authors.

3. *“Some of the data are either not quantified or not quantified appropriately. For instance, most panels in Fig. 2, the entirety of Fig. 3, and large portions of Fig. 4 lack quantitative analyses.”*

We thank Reviewer 2 for this comment. We have now quantified TRAP staining in the

premaxilla, mandible, and maxilla embryonically (please see Figure 2) and von Kossa staining areas in the premaxilla, mandible, and maxilla embryonically (please see Figure 3). We have also now included P1 measurement of the interparietal bone, occipital bone, malleus, stapes, and premaxilla for both CD1 (please see Figure 4) and C57 (please see Supplementary Figure 5) strains of mice. Together, 16 additional graphs displaying various forms of quantification, and stratified by sex, have been included. We would like to point out that we did not include any quantifications for the cranial base in Figure 4 as these structures are already quantified using microCT in Figure 5.

4. *“The presentation and quantification of male versus female data are inconsistent: in some bar graphs, male and female data appear to be pooled, whereas in others they are presented separately. This makes it difficult to assess the extent to which sex influences the observed phenotypes. A clear and consistent strategy for data stratification separating male and female data throughout with explicitly justified pooled analyses would enable rigorous evaluation of sex-specific and overall effects.”*

We thank Reviewer 2 for this comment and apologize for not making our prior decisions for pooling versus sex-stratification more obvious in our first submission. To be consistent across the manuscript and prevent differences in opinion with respect to when pooling is justified, we have stratified all our data by separating males and females for all 88 graphs in the manuscript, allowing more rigorous evaluation of sex-specific effects. To achieve this, additional experiments were performed across the study to increase our N's and allow for sex-stratification.

5. *“A minor comment is that although the Results section is divided into seven subsections, only Figs. 1 and 2 are used to support the first subsection, while all data corresponding to subsections 2 and 3 are placed entirely in the supplementary figures. In contrast, Figs. 3 and 4 are devoted solely to subsection 4. Perhaps this could be rebalanced to align the figures with the sections.”*

We thank Reviewer 2 for this comment. Although we were unable to adjust all the subsections in the results to provide more alignment with the figures, we did notice that in our prior submission subsection 4 of the results described findings from 4 Figures, while the other subsections only described findings from 1-2 Figures. Therefore, in the revised submission we have subdivided subsection 4 into 2 subsections, the first of which describes findings from 2 Figures (1 main Figure & 1 Supplemental Figure) and the second of which also describes findings from 2 Figures (1 main Figure & 1 Supplemental Figure).

Thank you for reviewing our manuscript, we look forward to receiving your advice on the suitability of our manuscript for Development.

Second decision letter

MS ID#: dev.205423R1

MS TITLE: CSF1R+ macrophage and osteoclast depletion impairs neural crest proliferation and craniofacial morphogenesis

AUTHORS: Felix Ma, Rose Ru Jing Zhou, Matthew Rosin, Iris Zhou, Sabrina Ownsworth, Rouzbeh Ostadsharif Memar, Vincent B. Wong and Jessica M. Rosin

Dear Dr Rosin,

I am happy to tell you that your manuscript has been accepted for publication in Development, pending our standard publication integrity checks.

Reviewer 1

Advance summary and potential significance to field

The authors have responded to most of my criticisms by removing the parts of the text that were incorrect and/or unnecessary. This editing has resulted in a greatly improved report of their interesting study. I have only a few more minor suggestions:

L121: The meaning of my comment was not fully understood. Please amend line 121 to read: “to be assessed by enhanced green fluorescent protein (EGFP+) signal”

L 314 and L318: change “unmineralized space” to “unmineralized areas”.

L605: “embryogenesis; albeit...” A semi-colon acts as a conjunction and must be followed by a clause with a verb in it. There are one or two other examples of this grammatical error.

L608: there is no direct tissue interface between the lens and the retina.

L624-626: “however, there is a paucity of literature focusing on craniofacial morphogenesis during the embryonic period (see Table S3)”. I was shocked by this statement - please add something to clarify that you are specifically referring to the mice with the genetic construct studied in this paper!

Reviewer 2

Advance summary and potential significance to field

The authors have done a nice job of addressing the reviews.